

**In-situ and Denuder Based Measurements of Elemental and Reactive**
**Gaseous Mercury with Analysis by Laser-Induced Fluorescence. Results**
**from the Reno Atmospheric Mercury Intercomparison Experiment.**
Anthony J. Hynes[*], Stephanie Everhart, Dieter Bauer, James Remeika, and Cheryl Tatum
Ernest[1]
Division of Marine and Atmospheric Chemistry, Rosenstiel School of Marine and
Atmospheric Science, University of Miami, 4600 Rickenbacker Causeway, Miami,
Florida 33149.
[1]current address:Atmospheric Chemistry Department, Max Planck Institute for
Chemistry, Hahn-Meitner-Weg 1, Mainz, 55128, Germany
[*] Corresponding author. Tel.: +1-305-421-4173; fax: +1-305-421-4689. E-mail address:
ahynes@rsmas.miami.edu (A. J. Hynes)
**Abstract**
The University of Miami (UM) deployed a sequential two photon laser-induced fluorescence (2P-LIF)
instrument for the in-situ measurement of gaseous elemental mercury, Hg(0), during the Reno Atmospheric
Mercury Intercomparison Experiment (RAMIX) campaign. A number of extended sampling experiments,
typically lasting 6-8 hours but on one occasion extending to ~24 hours, were conducted allowing the 2P-
LIF measurements of Hg(0) concentrations to be compared with two independently operated instruments
using gold amalgamation sampling coupled with Cold Vapor Atomic Fluorescence Spectroscopic (CVAFS)
analysis. At the highest temporal resolution, ~5 minute samples, the three instruments measured
concentrations that agreed to within 10-25%. Measurements of total gaseous mercury (TGM) were made by
using pyrolysis to convert total oxidized mercury (TOM) to Hg(0). TOM was then obtained by difference.
Variability in the ambient Hg(0) concentration limited our sensitivity for measurement of ambient TOM
using this approach. In addition, manually sampled KCl coated annular denuders were deployed and
analyzed using thermal dissociation coupled with single photon LIF detection of Hg(0). The TOM
measurements obtained were normally consistent with KCl denuder measurements obtained with two
Tekran speciation systems and with the manual KCl denuder measurements but with very large uncertainty.
They were typically lower than measurements reported by the University of Washington (UW) Detector for
Oxidized Hg Species (DOHGS) system. The ability of the 2P-LIF pyrolysis system to measure TGM was
demonstrated during one of the manifold HgBr$_2$ spikes but the results did not agree well with those reported
by the DOHGS system. The limitations of the RAMIX experiment and potential improvements that should
be implemented in any future mercury instrument intercomparison are discussed. We suggest that
instrumental artifacts make a substantial contribution to the discrepancies in the reported measurements





over the course of the RAMIX campaign. This suggests that caution should be used in drawing significant
implications for the atmospheric cycling of mercury from the RAMIX results.

**1.0 Introduction:**

The environmental and health impacts of mercury pollution are well recognized with impacts on

human health and broader environmental concerns (U.S. EPA., 2000; UNEP, 2013; Mergler et al. 2007;
Diez, 2009; Scheuhammer et al., 2007). There have been extensive reviews of global emissions,
measurements and biogeochemical cycling of mercury, (Mason, 2009; Streets et al., 2011; Pirrone et al.
2009; Lindberg et al., 2007; Ebinghaus et al., 2009; Sprovieri et al., 2010; Selin, 2009) The concerns
associated with the mercury problem have resulted in attempts to regulate and control emissions at both
national and international levels. The latest attempt in the United States is incorporated in the Mercury and
Air Toxics Standards (Houyoux, and Strum, 2011; US EPA, 2013)  and international efforts by the United
Nations Environment Program have led to the Minamata Convention on Mercury, a global
legally binding treaty on mercury controls (UNEP, 2008; UNEP, 2013; UNEP, 2014).
There is a reasonable consensus on typical background concentrations of atmospheric mercury, which are
extremely low. Currently concentrations range from 1.2–1.4 ng m$^{-3}$ in the Northern Hemisphere and 0.9–
1.2 ng m$^{-3}$ in the Southern Hemisphere and appear to be decreasing ( Slemr et al., 2011) [ 1 ng m$^{-3}$ is ~
$3 \times 10^6$ atoms cm$^{-3}$ or ~ 120 ppq (parts per quadrillion)].. Until recently it has been accepted that most of the
mercury found in the boundary layer is elemental mercury, Hg(0) (Lindberg et al., 2007). Oxidized or
reactive gaseous mercury (RGM), normally assumed to be in the Hg(II) oxidation state, has not been
chemically identified and is thought to constitute a very small fraction of the total mercury concentration
although recent work (Gustin et al., 2013, Ambrose et al., 2013) challenges this view. Our overall
understanding of the atmospheric chemistry of mercury and the detailed elementary chemical reactions that
oxidize Hg(0) is poor (Lin et al., 2006, Hynes et al., 2009; Subir et al., 2012) and the uncertainty of both
the chemical identity and measurements of speciated oxidized mercury places few constraints on models.
Atmospheric measurements of mercury represent a significant challenge in ultra-trace analytical chemistry
and the issues associated with current techniques have been discussed by Gustin and Jaffe (2010). We have
developed a laser-based sensor for the detection of Hg(0) using sequential two-photon laser-induced
fluorescence (2P-LIF) (Bauer et al., 2002; Bauer et al. 2014).  The instrument is capable of fast, in-situ,
measurement of Hg(0) at ambient levels. By incorporating pyrolysis to convert RGM and particulate
mercury to Hg(0) it is possible to measure total gas phase mercury (TGM) and hence to measure total
oxidized mercury (TOM, i.e. the sum of gas phase and particulate bound oxidized mercury) by difference.
The Reno Atmospheric Mercury Inter-comparison Experiment (RAMIX) offered an opportunity to deploy
the 2P-LIF instrument as part of an informal field intercomparison at the University of Nevada Agricultural
Experiment Station (Gustin et al., 2013, Ambrose et al., 2013; Finley et al., 2013). RAMIX was an attempt
to inter-compare new Hg measurement systems with two Tekran 2537/1130/1135 systems. This is the



instrumentation that is currently in use for the overwhelming majority of atmospheric Hg measurements.
Participants included the University of Washington (UW), University of Houston (UH), Desert Research
Institute (DRI), University of Nevada Reno (UNR) and the University of Miami (UM). The specific goals
for the project were:
1- Compare ambient measurements of gaseous elemental mercury, Hg(0), gaseous oxidized mercury
(RGM) and particulate bound mercury (PBM) by multiple groups for 4 weeks.
2- Examine the response of all systems to spikes of Hg(0) and $HgBr_2$.
3- Examine the response of all systems to Hg(0) in the presence of the potentially interfering
compounds ozone and water vapor.
4- Analyze the data to quantify the level of agreement and the results of interference and calibration
tests for each measurement system.
In practice the instrument operated by UH only measured Hg(0) for the first week of the campaign and the
cavity ring down spectroscopy (CRDS) instrument deployed by DRI did not produce any data. Hence
RAMIX was primarily an intercomparison of the UM 2P-LIF instrument, the UW Detector for Oxidized
Hg Species (DOHGS) that is based on two Tekran 2537 instruments, and a Tekran 2537 and two
2537/1130/1135 speciation systems deployed by UNR. Under these circumstances we were not able to
compare 2P-LIF measurements made at high temporal resolution with the CRDS instrument. It did allow us
to compare the 2P-LIF sensor with independently operated instruments that use preconcentration on gold
coupled with analysis by CVAFS and to examine potential interference effects. Our focus here is to
compare the short term variation in GEM on the timescale that the CVAFS instruments operate, ~ 5 minute
samples, and examine the ability of the different instruments to capture this variation. In addition, we made
measurements of TGM and hence TOM by difference and also employed manual denuder measurements to
attempt to measure RGM directly. In prior publications, Gustin et al. (2013) and Ambose et al. (2013)
provide their interpretation of the RAMIX results and their conclusions have very significant implications
for our understanding of atmospheric mercury chemistry. In this work we offer a contrasting view with
different conclusions.
**2.0 Experimental**
**2.1 RAMIX Intercomparison.** A detailed description of the RAMIX location and the local meteorology
was provided by Gustin et al. (2013). The original RAMIX proposal included participation from Tekran
Corporation to build and test a field-deployed, high-flow sampling manifold that could be reliably spiked
with 10-100 parts per quadrillion of RGM. Tekran proposed to supply both GOM and GEM spiking using
independent generators that were traceable to NIST standards and would be independent of the detection
systems being evaluated. However, due to time constraints Tekran believed that it was unlikely that the
manifold and ultra-trace spiking system could be manufactured and fully tested to their standards, so they
declined to participate in RAMIX (Prestbo, 2016). Instead, the UW group stepped in to supply and operate
the sampling manifold and spiking system and the details of its characterization are provided in Finley et al.
(2013). During the RAMIX campaign the 2P-LIF instrument sampled on 18 days, typically sampling for



between 4 and 6 hours. The longest period of continuous sampling lasted for 26 hours and occurred on
September 1st and 2nd. Over this 18 day period we sampled from the RAMIX manifold and, in addition, at
the end of the campaign we sampled ambient air independently and also attempted to measure TOM by
pyrolyzing the sample air and measuring the difference between Hg(0) and TGM. We also sampled RGM
using KCl coated annular denuders using LIF for real-time analysis.
**2.2 The 2P-LIF system**

Bauer et al. (2002, 2003, 2014) provide a detailed description of the 2P-LIF instrument including

the operating principles, linearity tests and examples of experimental data. In summary, the system uses
sequential two-photon excitation of two atomic transitions in Hg(0) followed by detection of blue shifted
LIF. The instrumental configuration at RAMIX utilized an initial excitation of the Hg $6^3P_1$-$6^1S_0$ transition
at 253.7 nm, followed by excitation to the $7^1S_0$ level via the $7^1S_0$ - $6^3P_1$ transition at 407.8 nm. Both
radiative decay and collisional energy transfer produce population in the $6^1P_1$ level. Blue shifted
fluorescence was then observed on the strong $6^1P_1$ - $6^1S_0$ transition at 184.9 nm using a solar blind
photomultiplier tube (PMT).  By using a solar blind tube that is insensitive to laser scatter at the excitation
wavelengths very high sensitivity is possible. The use of narrowband excitation of two atomic transitions
followed by detection of laser-induced fluorescence at a third wavelength precludes the detection of any
species other than Hg(0).  The 2P-LIF instrument requires calibration, so Hg(0) was also measured with a
Tekran 2537B using its internal permeation source as an absolute calibration. In prior field campaigns we
have been able to transport the Tekran with it remaining powered on which is important in maintaining the
stability of the permeation oven. This was not possible during the move from Miami to Reno and so the
Tekran was powered down for about one week prior to the start of measurements. We sampled from the
RAMIX manifold, which was below ambient pressure, through ~25 ft of ¼ in Teflon tubing.  The original
RAMIX plan called for all instruments to be located close to the manifold for optimal sampling.
Unfortunately the positioning of the trailers at the actual site precluded this and forced us to use a long
sampling line. As a result, the internal pump on our Tekran was not able to draw the 2 SLPM required for
sampling and an auxiliary pump was placed on the Tekran exhaust to boost the flow.
Under atmospheric conditions the 2P-LIF instrument cannot detect RGM so, in principle, this does not
need to be removed from the sample gas. However, deposition of RGM on the sampling lines followed by
heterogeneous reduction to GEM could produce measurement artifacts. The limit of detection for Hg(0)
during RAMIX was ~30 pg m$^{-3}$ for a 10 s or 100 shot average.

**2.3 Measurements of TGM and TOM**
We attempted to use the 2P-LIF instrument to measure TGM and hence TOM by difference. Although we
have routinely used this approach to convert $HgCl_2$ and $HgBr_2$ to Hg(0) in the laboratory, this was our first
attempt to measure total oxidized mercury at ambient  concentrations. A second sampling line was attached
to the RAMIX manifold and a pyrolyzer was located directly at the manifold sampling port.  The pyrolyzer
consisted of an ~0.6 cm o.d. quartz tube, 15 cm in length and partially filled with quartz wool.  Wrapped





Nichrome wire encompassed an 8 cm section of tube that was heated until the quartz began to glow. The
high temperature inside the pyrolyzer reduces both RGM and particulate mercury in the manifold air to
Hg(0), which is then monitored by 2P- LIF and gives the sum of oxidized (both gaseous and particulate)
and elemental mercury, i.e. TGM. Directly sampling from the manifold and measuring ambient Hg(0) then
allows the concentration of TOM to be calculated as the difference between the two signals.  Sampling was
therefore switched between the pyrolyzed and unpyrolyzed sample lines in, typically, 5 min intervals to
attempt to track fluctuations in [Hg(0)] that would obscure the relatively small signal increase attributable
to  TOM.
**2.4 Manual Denuder Sampling of RGM**

We conducted manual denuder sampling on seven afternoons during the RAMIX campaign to

attempt to quantify total RGM, We sampled using both KCl coated annular denuders and uncoated tubular
denuders that were then analyzed using programmable thermal dissociation (Ernest et al., 2013). In both
cases we monitored the Hg(0) that evolved during RGM decomposition, in real time using single photon
LIF. Only the annular denuder results are presented here. The use of denuder sampling coupled with
thermal dissociation has been described by Landis et al.(2003) and is used in the Tekran Model 1130
Mercury Speciation Units deployed during RAMIX. Air is pulled through a KCl coated annular denuder
which captures RGM but transmits elemental and particulate mercury. After a period of sampling, typically
one hour, the denuder is flushed with zero grade air and the denuder is heated to 500°C. The RGM is
thermally decomposed producing elemental mercury that desorbs from the denuder surface and is then
captured and analyzed by a Tekran 2537.  The KCl coated annular denuders used here were manufactured
by URG Corporation and were identical to those described by Landis et al for manual sampling. They were
located on top of one of the RAMIX instrument trailers a few feet from the entrance to the RAMIX
manifold inlet. The denuders sampled at 10 SLPM, they were not heated and the integrated
elutriator/acceleration jet and impactor/coupler described by Landis et al. and incorporated in the Model
1100 speciation unit were not placed on the denuder inlet. Hence no type of particle filtering was used on
the inlets. Prior to sampling, the denuders were cleaned by heating to 500 °C and then bagged and taken to
the sampling site. After a period of sampling that varied from ~1 to 4 hours, the denuders were capped,
placed in sealed plastic bags, and transported to the analysis lab at the University of Nevada, Reno. On
most of the sampling days a single denuder was opened and then immediately bagged serving as a field
blank. On the final two days of sampling, denuders were sampled in pairs, i.e with two denuders connected
inline so that the front denuder sampled RGM and the rear denuder served as a blank and monitor of bleed-
through of RGM. The blank concentrations are typically low as shown in Table 1: however on September
10[th] the blank shows a very high value that is indicative of significant contamination at some point during
the cleaning or sampling process.  For the analysis, a flow of He passed through the denuders and then into
a fluorescence cell where any Hg(0) in the flow was detected by LIF.  The LIF was monitored by two
PMT's set to different gains to increase the dynamic range of the detection system. Prior to the analysis, a
known amount of mercury was injected into the flow through a septum using a transfer syringe. The





syringe sampled from a Tekran Model 2505 Mercury Vapor Primary Calibration Unit. Without disrupting
the gas flow the denuder was then placed in a clamshell tube furnace that had been preheated to 500°C. The
evolution of the Hg(0) was monitored for, typically, 5-10 minutes and after the LIF signal had returned to
baseline a second calibration injection was performed.  A frequency doubled, Nd-Yag pumped dye laser
was used to excite the Hg(0) $6^3P_1$-$6^1S_0$ transition at 253.7 nm and resonance LIF was observed at the same
wavelength. In this approach, the detection PMT detects both LIF and laser scatter, hence sensitivity is
limited by the ratio of intensity of the LIF signal to the laser scatter. Since the $6^3P_1$ level is efficiently
quenched by both $O_2$ and $N_2$ (Breckenridge and Unemoto, 2007) the thermal analysis was performed in He
buffer gas to achieve good detection sensitivity.  The excitation beam then passed through a reference cell
that contained a steady flow of Hg(0) from a permeation source.  The LIF signal from the reference cell
served to confirm that the laser output was stable.
3.0 **Results:**
3.1 RAMIX Manifold
As noted above, the RAMIX manifold had to be constructed and tested by the UW group under tight time
constraints. A critique of the manifold performance has been presented by Prestbo (2014) and we detail
some key issues here. The manifold deployed at RAMIX was a different size than the prototype tested in
the laboratory.  The laboratory manifold showed very large variation in calculated transmission efficiencies
of GEM after spiking with a permeation source. Recoveries from 71-101% were reported for short-term
spikes. The GEM source used for spiking was calibrated by a Tekran 2537B. After the equipment was
moved to the RAMIX site the permeation tube output increased. The authors also acknowledge a
significant uncertainty (± 15%) in the RAMIX manifold flow measurements that were required to calculate
spike concentrations; hence this is the minimum uncertainty in calculated spike concentrations.

We find that several independent measurements of  GEM spikes differ by as much as 30% from

the value calculated by the manifold operators suggesting that (± 15%) underestimates the uncertainty.
Because of these considerations we believe the RAMIX manifold is best treated as a semi-quantitative
delivery system and it is most useful to focus on sampling periods when multiple independent instruments
show reasonable agreement.

**3.2 UM Tekran Performance**
In evaluating the first week of the UM RAMIX measurements it became clear that there was some non-
linearity in the relative responses of the 2P-LIF and UM Tekran systems and that better agreement was
obtained by referencing the Hg(0) concentration to the UNR Tekran. Gustin et al., (2013) concluded that
the UNR Tekran, based on the inlet configuration, only measured Hg(0) and they suggested that the UM
system, due to the long sampling line, was measuring TGM.  We compared the manifold Hg(0) readings
from the UM and UNR Tekrans over the first 260 hours in which we took measurements. The absolute
concentration difference relative to the UNR instrument is shown in Figure 1. Hour zero corresponds to 9
am on August 26th when we started measurements and hour 260 corresponds to midnight on September 5th.





221 Over the first 24 hours the UM Tekran is offset by ~0.5 ng m$^{-3}$ and then jumps to ~ 2 ng m$^{-3}$ at hour 30 on

222 August 27$^{th}$ with the difference decreasing over the next week of measurements in an almost linear fashion.

223 Over most of this period the UW Tekran did not report Hg(0) measurements other than a small set of

224 measurements on August 28$^{th}$ that are offset by ~0.5 ng m$^{-3}$ relative to the UNR Tekran.  It can be seen that

225 by hour 250 on September 5$^{th}$ all three instruments had converged. After this period the agreement between

226 the UNR and UM Tekrans was good until September 8$^{th}$, when the UM instrument became contaminated

227 after a malfunction of our permeation oven, requiring replacement with a backup Tekran 2537A unit. We

228 conclude that the difference between the UM and UNR instruments is likely to be an experimental artifact

229 possibly associated with the fact that the UM instrument had been powered down for almost one week and

230 relocated to a site at a significantly different ambient pressure. The initial abrupt change to a large offset

231 followed by the linear decrease over 300 hours cannot, in our view, be caused by any type of chemistry

232 within the manifold, nor can it be indicative of the UM instrument measuring TGM rather than Hg(0).

234 **3.3 2P-LIF Measurements**

235 The absolute Hg(0) concentrations reported for the 2P-LIF measurements typically use a single 10-minute

236 section of Tekran concentration data to calibrate the 2P-LIF signal and place it on an absolute concentration

237 scale. The complete time series of measurements then gives a long-term comparison of the 2P-LIF and

238 Tekran instrumentation with the absolute 2P-LIF concentrations based on the single 10-minute calibration

239 point.

241 **3.3.1 September 5$^{th}$**

242   This was the first occasion on which the three independent Tekran 2537 instruments and the 2P-

243 LIF system reported simultaneous measurements. The 2P-LIF system sampled from the RAMIX manifold

244 for approximately 6.5 hours from ~10:30 am to 5 pm. Over the course of the sampling period there were

245 two spikes of Hg(0) lasting one and two hours, respectively.  The UW manifold team reported an initial 10

246 am Hg(0) spike concentration of 26.5 ng m$^{-3}$ dropping to 24.4 ng m$^{-3}$ over the course of the one hour spike.

247 The two hour spike that began at 1 pm was reported to be ~12.4 ng m$^{-3}$ dropping to 10.5 ng m$^{-3}$ over the

248 course of two hours.  The ambient airflow in the manifold was spiked with HgBr$_2$ for the whole of this

249 sampling period and the reported level of the HgBr$_2$ spike varied between 0.6-0.7 ng m$^{-3}$.  The levels of

250 HgBr$_2$ measured by the DOHGS instrument were consistent with this but the concentrations reported by the

251 UNR speciation units were considerably lower and with a significant discrepancy between the two

252 speciation units.  Figure 2a shows the sequence of Hg(0)  measurements from the UNR, UW and UM

253 Tekrans together with the 5 minute averages of the 2P-LIF signal. The 2P-LIF instrument began manifold

254 measurements in the middle of the initial 10 am Hg(0)  spike and is scaled to the concentration at this time

255 which all three Tekrans measured as ~22.5 ng m$^{-3}$.  The three Tekrans agree to better than 5% during both

256 of the manifold spikes and, based on a pre-spike ambient concentration of 2 ng m$^{-3}$ it suggests that the

257 initial spike concentration was ~ 20.5 ng m$^{-3}$. This suggests that the reported spike concentration was ~25-



30% larger than the actual concentration introduced into the manifold. Fig. 2b shows an expanded
concentration scale to highlight the nominally ambient measurements. There is some suggestion that it took
some time for the spike to be completely removed, particularly after the second spike. At the completion of
the second spike all the instruments drop to ambient but the UNR instrument sees two Hg(0) "pulses".
Interestingly these show up with greatly reduced amplitudes in the UW and UM Tekran signals and also in
the 2P-LIF signal. Figure 3 shows the % difference of the other instruments relative to the UM Tekran and
over most of the sampling period the agreement between all the measurements is better than 10% over an ~
7 hour period with 5 minute sampling resolution.  This indicates that the 2P-LIF instrument is capable of
stable operation over an extended time period with any drifts being corrected by normalization to the
reference cell. Well calibrated independently operated Tekrans should be capable of agreement to better
than 5% based on tests performed by the manufacturer and this level of agreement is achieved during
subsets of the sampling period. It is not clear if the deviations that are observed, particularly the large
deviations seen by the UNR Tekran after the second spike are related to presence of elevated levels of
$HgBr_2$, or other issues related to manifold operation. The fact that all the instruments observed these Hg(0)
pulses suggests that the artifact may be related to a process in the manifold rather than in in the UNR
sampling line. However the significant differences in the magnitude of Hg(0)  pulses observed by the
different instruments are difficult to rationalize.

**3.3.2 September 1st and 2nd**
The UM and UNR systems sampled simultaneously for a 22 hour period offering an opportunity to
compare the instruments over an extended sampling period. This sampling also occurred prior to any of the
manifold spikes that introduced substantial concentrations of $HgBr_2$ into the manifold and sampling lines.
Unfortunately, the UW instrument did not report any measurements during this sampling period. The UM
system sampled for 26 hours and the complete dataset is described elsewhere, (Bauer et al. 2014). This
includes a detailed analysis of the short-term, i.e. 1-10 seconds, variation in the Hg(0) concentration and the
ability of the 2P-LIF system to capture this. Here we focus on the simultaneous sampling period and the
variability that should be resolvable by both of the Tekrans and the 2P-LIF instruments. SI Figure 1 shows
the 24 hour sampling period with the 2P-LIF signal calibrated by the UM Tekran concentration at the
beginning of hour 13 (i.e 1 pm on September 1st) and the corresponding measurements from the UNR
Tekran. SI Figure 2 shows the same data with an expanded y-axis to highlight the variation in the ambient
measurements. All three instruments track each other quite well over the first 10 hours and then measure a
nocturnal increase in Hg(0) which shows greater medium term variability in the concentration. The 2P-LIF
concentrations are approximately 20% greater than the Tekran measurements during this period. At hour 33
(i.e. 9 am on September 2nd) there was a manifold spike with a reported concentration of 12.9 ng m$^{-3}$
dropping to 11.9 ng m$^{-3}$ over the course of one hour.  The UNR Tekran is ~6% lower, the UM Tekran is
~20% lower and the 2P-LIF ~22% higher than the calculated spike concentration. SI Figure 3 shows the
same measurement set but with all instruments normalized to the second manifold spike at hour 33. Figure





4 shows an expanded y-axis, the concentration scale, focusing on the ambient concentration measurements.
It is apparent that we now see better agreement between the 2P-LIF and the UNR Tekran but that the UM
Tekran lies systematically higher than the UNR Tekran. Figure 5 shows a three hour subset of the
measurements corresponding to 5-8 am on the morning of September $2^{nd}$. The variation between the
instruments is greater than 5% and the short term variations in the Hg(0) concentration vary between the
three instruments. Using either calibration approach we see that all instruments capture both the nocturnal
increase in Hg(0) concentration and the greater variability in the signal but that there are differences in the
amplitude of the variability.

**3.3.3 Hg(0) Intercomparison Conclusions**
Almost all of the measurements of atmospheric concentrations of Hg(0) have been made with
CVAFS instrumentation and the majority of those measurements have utilized the Tekran 2537. This work
provides the first extensive comparison of the Tekran 2537 with an  instrument that is capable of fast in-situ
detection of Hg(0) using a completely different measurement technique. Measurements over two extended
sampling periods show substantial agreement between the 2P-LIF and Tekran measurements and suggest
that all the instruments are primarily measuring the same species. Intercomparison precision of better than
25% was achievable over an extended sampling period and precision of better than 10% was achieved for
subsets of the sampling period. As we discuss below it is difficult to determine the extent to which
interferences from GOM contribute to the differences observed.

**3.4 Interference Tests.**
As noted above, one component of the initial RAMIX proposal was an examination of the response of the
various sensors to potential interfering compounds $HgBr_2$, $O_3$ and $H_2O$.  An analysis of the 2P-LIF
detection approach suggests that, at the spike levels employed during the RAMIX campaign, neither $HgBr_2$
nor $O_3$ should have any interference effects. Changes in the concentration of $H_2O$ do affect the 2P-LIF
signal because $H_2O$ absorbs the 2P-LIF fluorescence signal and may quench the fluorescence. In addition,
$O_2$ also absorbs the 2P-LIF signal and quenches fluorescence thus a change in the $O_2$ concentration will
affect the linearity of the response.  We have presented a detailed discussion of these effects (Bauer et al.,
2014) including an examination of two types of interferences that have been observed in LIF sensors
applied in atmospheric and combustion environments and concluded that these are not potential problems
in 2P-LIF measurements of atmospheric Hg(0). As we have noted previously (Bauer et al., 2014),
condensation in our sampling lines can produce artifacts in Hg(0) concentration measurements. Because of
the low humidity in Reno it was not necessary to use any type of cold trap during ambient measurements
but we did use a trap during manifold spikes of $H_2O$ so our measurements do not address this as a potential
interference.
**3.4.1 $O_3$ Interference Tests.**



On September 7[th] an ozone interference test was conducted by simultaneously spiking the

sampling manifold with a high concentrations of Hg(0) and ozone. The spike in Hg(0) lasted from 9am to
7:30 pm and there were two ozone spikes, each of two hours duration. A comparison of the UM, UW and
UNR Tekrans and the 2P-LIF signal is shown in Figure 6. The UW Tekran only measured for a portion of
this period but agrees reasonably well with the other Tekrans. The 2P-LF signal is calibrated by the UM
Tekran reading during the initial Hg(0) spike at hour 9.30. The 2P-LIF signal was online for 6 minutes at
the beginning of the first ozone spike and then went offline for ~40 minutes for instrument adjustments.
When the 2P-LIF came back online the magnitude of the normalized signal was low relative to the Tekrans.
At hour 13 all three instruments converge and agree well over the course of the second spike. The
magnitude of the 2P-LIF signal could have been affected adversely by the adjustments but any reduction in
signal should have been compensated by a corresponding change in the reference cell. The elevated levels
of ozone were introduced into the manifold by UV irradiation of $O_2$ and adding the $O_2/O_3$ gas mixture
directly into the manifold produced a reported ~8% relative increase of $O_2$ levels in the manifold mixing
ratio. As we note above this additional $O_2$ would absorb some of the 2P-LIF signal but this would be a very
small effect. The enhanced quenching by $O_2$ is more difficult to assess but cannot explain the discrepancy
between the Tekrans and the 2P-LIF signal. In addition the agreement during the second ozone spike was
good. One possible explanation is that the increase in the $O_2$ mixing ratio was larger than calculated for the
first spike. A second series of $O_3$ spikes were conducted on September 13[th] when we were attempting to
measure total gaseous mercury using pyrolysis as described below. The 2P-LIF measurements switched on
a five-minute cycle between a pyrolyzed line that would have decomposed all the ozone in the sample and
a line containing the ambient air spiked with ozone. There was no difference in the 2P-LIF signal from the
two sampling channels again suggesting that $O_3$ has no interference effects.  The changes in the Hg(0)
concentration measurements track the predicted changes in calculated  spike concentration. However the
calculated spike concentration is 20-40% higher than the actual measurements obtained by the UM Tekran.

**3.5 Measurements of TGM and TOM**
We made attempts to use the 2P-LIF instrument to measure TGM and hence TOM by difference by
sampling through two manifold lines. A pyrolyzer was located at the manifold on one of the sampling lines
to measure TGM. The other sampling line measured ambient Hg(0). TOM was calculated from the
difference in the TGM and Hg(0) concentrations and in this sampling configuration the limit of detection
for TOM depends on the short term variability in ambient Hg(0) which is significant and shows a diurnal
variation.  The pyrolysis system was set up and tested on September 12. Manifold sampling was conducted
on the 13[th] and 14[th] and sampling from the trailer roof occurred on the 15[th]. We calculated the means of the
pyrolysis and ambient channel concentrations, and the difference which gives the TOM concentration. We
also calculated the standard deviations and standard errors (SE) and used these errors to calculate in
quadrature the 2SE uncertainty in the derived TOM concentration. However, as discussed below, the errors
in the means do not appear to capture the full variability in Hg(0), particularly at shorter sampling times.



### 3.5.1 September 14th


Our most extensive sampling took place on the 14th and we were able to sample for three ~ 2 hour periods
between 9 am and 8 pm. On this day there were multiple manifold spikes of $HgBr_2$ and also an $Hg(0)$ spike
and we have a made a detailed analysis of the data for each sampling period.
The third sampling period which included a large $HgBr_2$ spike provided the only definitive
opportunity to demonstrate the capability of 2P-LIF coupled with pyrolysis to measure oxidized mercury.
The third sampling period began at ~ hour 17.3 during a manifold $HgBr_2$ spike that began at hour 17. A
short $Hg(0)$ spike was also introduced at hour 18. Fig. 7 shows the 2P-LIF signals from the ambient and
pyrolyzed sampling lines together with the means and 1 standard deviation. The UM Tekran was offline at
this time and so the 2P-LIF concentrations are calibrated by the concentrations reported by the UNR
Tekran at the beginning of the $Hg(0)$ spike which are also shown. Both the UNR Tekran and UW Tekran
report very similar $Hg(0)$ concentrations during the $Hg(0)$ spike. Both systems report an $Hg(0)$
concentration of 6.7 ng m$^{-3}$ at the beginning of the spike which, since the pre-spike concentration was ~1.9
ng m$^{-3}$, corresponds to a spike concentration of 4.8 ng m$^{-3}$. This is lower than the calculated spike
concentration of 6.1 ng m$^{-3}$ reported by the manifold operators and suggests that the calculated spike was
~27% higher than the actual spike concentration introduced into the manifold. Fig. 8 shows the means of
each set of ambient and pyrolyzed measurements together with the 2σ variation and 2SE of the mean. Fig. 9
shows the TOM concentrations calculated from the difference together with 2SE in the TOM concentration.
The reported spike concentrations and DOHGS measurements are also shown. During the initial sampling
period between ~17.3- 17.8 hours the 2P-LIF pyrolysis measurements do not show evidence for an $HgBr_2$
spike. Taking the difference between the ambient and pyrolyzed measurements during this period we obtain
[TOM] = 0.05±0.05 ng m$^{-3}$.  Shortly before the introduction of the $Hg(0)$ spike we see clear evidence for an
increase in the $Hg(0)$ concentration in the pyrolysis sample relative to the ambient sample.  We speculate
that the manifold adjustments that were made to introduce the additional $Hg(0)$ spike produced either a
change in the flow or some other change in the manifold conditions that allowed the $HgBr_2$ spike to reach
our pyrolyzer, which, as mentioned above, was located at the manifold. This difference between the two
2P-LIF signals is clearly evident by inspection of Fig.7. The TOM concentration which should consist
almost exclusively of $HgBr_2$ is significantly larger than both the reported $HgBr_2$ spike concentration and the
concentrations reported by the DOHGS system which are in perfect agreement. Taking the difference
between the ambient and pyrolyzed measurements for hour 18.02-18.35 we obtain [TOM] = 1.20±0.17 ng
m$^{-3}$ with 2SE uncertainty. It is important to note again that the calculated $Hg(0)$ concentration is 27% larger
than the measured concentration. This large difference is most likely due to errors in the flows or the
permeation source output but it suggests significant uncertainty in the calculated concentration of the
$HgBr_2$ spike. In addition, it is clear that the DOHGS measurements show a different temporal profile of
TOM. The DOHGS system reports TOM concentrations that agree almost exactly with the calculated spike
concentration, at the beginning of the spike period and drop to a very low background level that is below
the detection limit at the end of the reported spike period. In contrast, the 2P-LIF measurements do not



show an increased TOM concentration until shortly before the introduction of the Hg(0) spike and they take
~20 minutes to drop to background levels. The UNR speciation systems sample for 1 hour and this is
followed by a 1 hour analysis period so they produce a single hourly average every two hours. During this
period the UNR speciation system Spec1 sampled for ~ 20 minutes during the spike period and then for a
further 40 minutes. Spec2 was sampling ambient air outside the manifold.
SI Figure 4 shows the 7s average of the 2P-LIF signal from the ambient and pyrolysis sample lines
for the first sampling period 8-10.45 hours together with the mean and 1standard deviation ( 1σ) variation
in the 2P-LIF signals. SI 5 shows the means together with the 2σ variation and 2SE of the mean. It is clear
that there is significant short term variability in the ambient Hg(0) concentration. SI Fig. 6 shows the TOM
concentrations calculated from the difference between the pyrolyzed and ambient channels together with
the calculated 2SE in the TOM concentration. The reported spike concentration and DOHGS concentration
measurements are also shown. If we take the means of the 2P-LIF ambient and pyrolysis measurements
during the reported spike period we obtain: ambient: $2.06\pm0.05$ ng m$^{-3}$ and pyrolyzed: $2.21\pm0.03$ ng m$^{-3}$
giving a TOM concentration of $0.145\pm0.05$ ng m$^{-3}$. The 2P-LIF measurements are consistent with the
detection of TOM but they are much lower than the calculated spike and DOHGS measurements.
SI Figs.7-9 show the corresponding plots for the second sampling period from ~ 12.2-14 hours.
The alternating sampling between the ambient and pyrolysis channels is more even and SI Fig. 7 shows that
there is still variability in ambient Hg(0). The means of all the samples give: ambient: $1.72\pm0.02$ ng m$^{-3}$,
pyrolyzed: $1.70 \pm 0.02$ ng m$^{-3}$. If we take the subset of measurements that coincide with the reported spike
we obtain: ambient: $1.79\pm0.02$ ng m$^{-3}$   pyrolyzed $1.77\pm0.02$ ng m$^{-3}$. In this case, the 2P-LIF measurements
do not detect $HgBr_2$ and are not consistent with the reported spike or DOHGS measurements.
SI Figs. 10 and 11 show the averages of the TOM concentrations from the 2P-LIF system together
with the measurements from the UNR speciation systems, the reported spike concentrations and 5 min
DOHGS concentrations. During this sampling period Spec1 sampled from the RAMIX manifold while
Spec2 sampled ambient air outside the manifold. Gustin et al.(2013) detail problems with the response of
the Spec2 system and applied a 70% correction that is also shown as "Spec2 corrected". Because both the
DOHGS and 2P-LIF pyrolysis systems are expected to measure the sum of gaseous (RGM) and particulate
(PBM) oxidized mercury we have plotted the sum of the RGM and PBM concentrations from the
speciation systems. They are plotted at the mid-point of the 1 hour sampling period.
Over most of the measurement period the 2P-LIF pyrolysis and Spec1 measurements are
consistent and lower than the DOHGS measurements. The exception is the large spike in TOM seen by the
2P-LIF system at hour 18. The spike occurred during the initial portion of Spec1 sampling and, although it
measures an increase in RGM relative to Spec2, the magnitude is not consistent with the 2P-LIF pyrolysis
observations.

**3.5.2 September 13[th]**



September 13$^{th}$ was the first day we were able to sample with the pyrolysis system and we sampled over a
period of 5 hours. The only manifold spike during this period was an O$_3$ spike at 1pm that lasted one hour
so the speciation instruments were attempting to measure ambient RGM.  SI Figure 12 shows averages of
TOM concentrations as measured by the 2P-LIF pyrolysis system together with the hourly averages as
measured by  DOHGS  and  UNR speciation instruments. The x-axis error shows the duration of the 2P-LIF
measurements together with 2SE y-axis error bars. Two of the averages of the 2P-LIF measurement give a
physically unrealistic negative concentration suggesting that combining the 2SE errors in the means of the
ambient and pyrolyzed channels underestimates the uncertainty in the TOM measurement.
**3.5.3 September 15$^{th}$.**

On September 15$^{th}$ we sampled from the trailer roof using the same sampling lines and again

alternating between the pyrolyzed and unpyrolyzed channels.  SI Figure 13 shows the averages of the 2P-
LIF signal from the ambient and pyrolysis channels together with the concentrations measured by the
Spec2 system that was sampling ambient air outside the manifold. The concentration obtained from the UM
denuder samples described below are also shown. The UW DOHGS and Spec1 systems were sampling
from the RAMIX manifold with continuous HgBr$_2$ spiking during this period. We see some evidence for
measurable RGM in the first hour of the measurements and this is not seen by Spec 2. Later measurements
show no evidence for measurable RGM concentrations.
**3.6 Limits of 2P-LIF detection of TOM**

As we have noted above, the limit of our detection of TOM depends on the short term variability

in the ambient Hg(0) concentration because we use a single fluorescence cell and switch between pyrolysis
and ambient channels. We have attempted to give an estimate of the uncertainty by taking two standard
errors of the means and combining the errors in quadrature to get an estimate of the uncertainty in the TOM
concentration. If the mean of the ambient Hg(0) concentration is not fluctuating significantly on the
timescale of channel switching this approach should give an accurate estimate of the uncertainty in TOM.
In fact our Hg(0) observations show that the fluctuations in the Hg(0) concentration show a significant
diurnal variation, with large fluctuations at night, decreasing over the course of morning hours and being
smallest in the afternoon. This can be seen in the long term sampling from September 1$^{st}$ and 2$^{nd}$ and in the
observations from September 14$^{th}$. The observation of statistically significant but physically unrealistic
negative TOM concentrations on September 13$^{th}$ may be explained by this. Such an artifact could be
produced by contamination in the Teflon valve switching system that alternates the flow to the fluorescence
cell. This type of contamination should produce a constant bias that is not actually observed. It appears that
the short term variability in Hg(0) concentration produces a small bias in some cases that is not averaged
out by switching between the ambient and pyrolyzed channels. For example on September 13$^{th}$ the initial
sample period of 1.2 hours gives an RGM concentration of  0.06±0.10 ng m$^{-3}$  while two shorter sampling
periods at hour 10.5 (36 min sample) and 13.5 (12 min sample) give 0.15±0.09 ng m$^{-3}$. Our results suggest
that the use of single detection channel with switching between ambient and pyrolyzed samples is not
adequate to resolve the small concentration differences that are necessary to be able to monitor ambient



TOM. It is necessary to set up two detection systems, one continuously monitoring ambient Hg(0) and the
other continuously monitoring a pyrolyzed sample stream giving TGM, to get the precision necessary to
monitor ambient TOM. Over most of the measurement periods our results are consistent with the lower
TOM values reported by the UNR speciation instruments although there is a large uncertainty in the
concentrations that is actually difficult to quantify. In addition, it is important to emphasize that this was
our first attempt to use the pyrolysis approach to attempt to measure TOM. It is possible that the pyrolyzer
was not working efficiently on September 13th. The results from September 14th are more difficult to
rationalize. The 2P-LIF pyrolysis system has the sensitivity to detect the much higher values of RGM
reported by the DOHGS system and the reported spike concentrations of $HgBr_2$. At higher concentrations,
as shown in Fig. 9, the 2P-LIF system can monitor $HgBr_2$ with ~10 minute time resolution. Our results,
however, cannot be reconciled with those reported by the DOHGS system or the spike concentrations
reported by the UW manifold team.

**3.7 Manual Denuder Measurements:**
As we describe above, our use of manual denuders is similar to that described by Landis et al. (2002) with
the exception that we do not incorporate the integrated elutriator/acceleration jet and impactor/coupler on
the denuder inlet. Feng et al. (2004) suggested that such impactors could reduce the efficiency of RGM
collection. Hence no type of particle filtering was used on the inlets. In addition, we used single photon LIF
to monitor the evolution of Hg(0) in real-time as the RGM decomposed on the hot denuder surface during
oven analysis. The analysis was carried out in He buffer gas and the Hg(0) concentration was calibrated by
manual injections. The first series of measurements, i.e. September 6-14th involved single denuder
sampling. On the 15 and 16th we employed tandem sampling with two denuders in series to assess the
extent of RGM "bleedthrough". We used two sets of denuders on the 15th and four sets of denuders on the
16th. Fig. 10 shows the raw data for a denuder analysis showing the preheat Hg(0) calibration injections and
the temporal profile of the Hg(0) LIF signal for one of the September 16th samples, denuder 1. The two
traces correspond to the two monitoring PMTs set at different gains to increase the dynamic range of the
measurements. Fig. 11 shows the calibrated profile for the same denuder together with the "blank" i.e. the
trailing denuder. The complete set of manual denuder data together with corresponding values for the UNR
speciation units that are closest in sampling time are shown in Table 1. Sampling occurred on denuders 1,
4, 6 and 7. The "trailing" denuders which we have treated as blanks, are denuders 3, 5, 8 and 9. The
advantage of monitoring the RGM decomposition in real-time is shown in the September 16th data. The
temporal decomposition profiles (TDP) for three of the denuders shown in Fig 11 and SI Figures 14 and 15
show reasonable agreement both in absolute concentration of Hg(0) and the time for decomposition to
occur. The fourth denuder sample, SI Fig. 16, is a factor of 4-5 higher in concentration and decomposes on
a longer time scale with significant structure in the TDP. Comparing the TDPs for all eight denuders it is
clear that the TDP for denuder 7, which shows the anomalously high value, is very different from the TPDs
for the other three sample denuders. We believe that this TDP is associated with particulate mercury that



has impacted on the denuder wall and decomposes on a slower timescale giving a very different temporal
profile from RGM that was deposited on the denuder wall. SI Table 1 shows the values of RGM obtained
from denuder analysis together with an indication of impact from a PBM component. We have also
included measurements from the UNR speciation systems that overlap with, or are close to, the times when
our measurements were made. We draw several conclusions from the measurements. The values we obtain
from simultaneous measurements that are not influenced by the presence of PBM agree reasonably well
with each other, are broadly consistent with the values reported by the Tekran speciation systems and are
typically much lower than the values from the UW DOHGS system. Two sets of tandem denuder
measurements from September 15 and 16 indicate that there is not a significant level of "bleedthrough"
onto the trailing denuders. This suggests that the large differences between the DOHGS system and the
UNR speciation systems are not due to specific problems with the RAMIX manifold or the speciation
systems deployed at RAMIX even though Spec 2 was not functioning properly as documented by Gustin et
al. (2013).  The tandem sampling also demonstrates that any denuder artifact is not a result of some type of
"bleedthough" artifact that is preventing RGM from being quantitatively captured by the first denuder.  It is
also noteworthy that the manually sampled denuders were at ambient temperature in contrast to the
speciation denuders that are held at 50 C. Hence the absolute sampling humidities are similar but the
relative humidities are very different. Finally, we suggest that there is value in monitoring RGM
decomposition in real time as diagnostic of particulate impact when utilizing the annular denuders without
the impactor inlet designed to remove  coarse particulate matter that may be retained due to gravitational
settling
**4.0 Implications of RAMIX results.**

We think a realistic assessment of the RAMIX results is imperative because the interpretation of

the RAMIX data and the conclusions presented by Gustin et al. (2013) and Ambrose et al. (2013) have
enormous implications for both our understanding of current experimental approaches to atmospheric
sampling of mercury species and to the chemistry itself. Speciation systems using KCl denuder sampling
are widely used in mercury monitoring networks worldwide to measure RGM concentrations and the
Gustin et al. (2013) and Ambose et al. (2013) papers  suggests these results greatly underestimate RGM
concentrations with no clear way to assess the degree of  bias.
**4.1 Intercomparison of Hg(0)**

The assessment of the Hg(0) measurements is a little different in the two manuscripts with

Ambrose et al. (2013), noting that "comparisons between the DOHGS and participating Hg instruments
demonstrate good agreement for GEM" they found a mean spike recovery of 86% for the DOHGS
measurements of GEM, based on comparisons between measured and calculated spike concentrations.
Gustin et al. (2013) suggest that the UM Tekran agreed well with measurements of TM reported by the
DOHGS system and they "hypothesize that the long exposed Teflon line connected to the UM Tekran unit
provided a setting that promoted conversion of RM to GEM, or that RM was transported efficiently through
this line and quantified by the Tekran system. The latter seems unlikely given the system configuration…",



where RM refers to reactive mercury. As we note above, we believe that the best explanation for
discrepancies between the UM and UNR Tekrans is an experimental issue with the UM Tekran response
during the initial period of sampling. We would suggest that data from September 5[th], one of the few
occasions when data from multiple instruments agreed over an extended period is not compatible with
either transmission or inline reduction of RGM in our sampling line. What is also significant from this data
is the very large discrepancy between the spike concentrations as measured independently by three
different Tekran systems and confirmed by the relative response of the 2P-LIF measurements and the
calculated spike concentration. The discrepancy, on the order of 25-30%, is larger than the manifold
uncertainties suggested by Finley et al. (2013). We note other examples of the measured Hg(0) spikes
being significantly lower than the calculated concentrations. In prior work we have shown that both the
Tekran and 2P-LIF systems show excellent agreement over more than 3 orders of magnitude in
concentration when monitoring the variation in Hg(0) in an $N_2$ diluent. It is to be expected therefore that the
"recovery" of high concentration spikes should show good agreement between the different instruments as
observed in the September 5[th] data. The difference between the observations and the calculated manifold
spike concentrations is, we would suggest, a reflection of the significant uncertainty in the calculated
manifold spike concentration and is not a reflection of reactive chemistry removing Hg(0). In addition,
random uncertainties in the flow calculations should not produce a consistently low bias relative to the
calculated spike concentrations. As we note above in section 3.1 Ambrose et al. report an increase in the
output of their Hg(0) permeation tube after the move to the RAMIX site but this assumes that their Tekran
calibration is accurate. The results are consistent with their Tekran measuring too high an output from the
permeation device. This is significant if the same Tekran is being used to calibrate the output of the $HgBr_2$.

A more difficult issue is the question of resolving the differences in the temporal variation of

ambient Hg(0) at the 5 minute timescale as captured by the different instruments. The Tekran systems
should be in agreement with a precision of better than 5% and the 2P-LIF system, with a much faster
temporal resolution and detection limit, should be capable of matching this. The differences here are not
consistently associated with a single instrument with, for example, the 2P-LIF having some systematic
offset with respect to the CVAFS systems. The extent to which the larger (i.e. larger than 5%) observed
discrepancy which ranged from 10% to 25% is a result of interferences or simply a reflection of instrument
precision is difficult to assess. We note again that the UM instruments had to sample through a very long
sampling line and we expect that oxidized mercury is deposited on the sampling line. However it is not
possible to assess the extent to which oxidized mercury is reduced back to its elemental form introducing
small artifacts. As we suggest below, an intercomparison of instrument response to variation in Hg(0)
concentrations in a pure $N_2$ diluent with the Hg(0) concentration varying between 1-3 ng m$^{-3}$ would provide
a definitive baseline measurement of the instrument intercomparison precision and accuracy. We suggest
that such a measurement is a critical component of any future intercomparrison of mercury instrumentation.

**4.2 Comparison of Total Oxidized Mercury**



To the best of our knowledge RAMIX is the only experiment that has measured ambient TOM using
multiple independent techniques. It should again be emphasized that the TOM measurements using
pyrolysis with 2P-LIF detection were the first attempt to perform such measurements and the use of a
single channel detection system introduced large uncertainties into the measurements. The very large
discrepancies between the measurements of TOM reported by the DOHGS system, the Tekran speciation
systems and the limited number of 2P-LIF pyrolyzer measurements are the most problematic aspect of the
RAMIX measurement suite. Work prior to RAMIX  and suggested a potential ozone and/or humidity
interference in the operation of KCl coated annular denuders and a number of studies since have also
reported such an effect (Lyman et al., 2010; McClure et al., 2014).  Typically however the differences
between the RAMIX measurements are large and are not germane to the differences between the DOHGS
and 2P-LIF pyrolyzer measurements. The SI Figures give an example of the differences between the
DOHGS measurements and the denuder and 2P-LIF measurements. Ambose et al. (2013) note that the
DOHGS measurements were, on average, 3.5 times larger than those reported by the Spec1 system and
summarize the comparison with denuder measurements as follows: "These comparisons demonstrate that
the DOHGS instrument usually measured RM concentrations that were much higher than, and weakly
correlated with those measured by the Tekran Hg speciation systems, both in ambient air and during $HgBr_2$
spiking tests." The discrepancy of a factor of 3.5 is an average value but, for example, examining the
September 14 data at ~5 am the DOHGS system is measuring in excess of 500 pg m$^{-3}$ compared with ~20
pg m$^{-3}$ measured by the speciation systems, a factor of 25 difference. At this point the Hg(0) concentration
was ~ 3 ng m$^{-3}$ so based on the DOHGS measurements oxidized mercury is ~ 15% of the total mercury
concentration. A recent study by McClure et al. (2014) provided a quantitative assessment of the extent to
which ozone and humidity impact the recovery of $HgBr_2$ on KCl recovery. They note that although they
provide a recovery equation to compare with other studies, they do not recommend use of this equation to
correct ambient data until more calibration results become available. In Fig 12, we show the ozone
concentration and absolute humidity for a 35 hour sampling period on September 13$^{th}$ and 14$^{th}$ that included
two ozone spikes and only sampled ambient TOM. Fig 13 shows the expected denuder recovery based on
the formula determined by McClure et al. which varies between a typical value of ~70% dropping to ~50%
during the ozone spikes. The figure also shows the reported recoveries i.e. the ratio of RGM as measured
by either the UNR speciation systems or the 2P-LIF system divided by the value reported by the DOHGS
system. These values are typically much lower than those predicted by the McClure recovery expression. In
addition, on September 13$^{th}$ and for most of the 14$^{th}$ the 2P-LIF pyrolysis system sees little or no evidence
for high spike concentrations of $HgBr_2$ but records levels that fluctuate around those reported by the
speciation systems. The one exception is the spike at hour 18 on September 14$^{th}$.
We suggest that the ability of the 2P-LIF pyrolysis system to monitor large spike concentrations is
shown by the measurements during the September 14$^{th}$ $HgBr_2$ spike at hour 18. The evidence for an
enhancement in the pyrolyzed sample stream is observable in the raw 7s averaged data and becomes clear
taking 5 minute averages. The absolute value of the pyrolyzed enhancement is obtained relative to the



626 concentration of the Hg(0) during the spike taken from the measurements by the UNR Tekran that are in

627 excellent agreement with the DOHGS Hg(0) values. The 2P-LIF measurements show a significantly larger

628 $HgBr_2$ concentration and a different temporal profile compared with the DOHGS instrument. In particular,

629 it is very difficult to rationalize the difference between the 2P-LIF and DOHGS systems during the first

630 hour of the spike. We would suggest it is difficult to make the case that both instruments are measuring the

631 same species. It is clear that the 2P-LIF pyrolyzer is operating efficiently based on the clear observation of

632 TOM at the end of the spike. We again note that the 2P-LIF system is not sensitive to TOM. It is important

633 to note that the DOHGS instrument requires an inline RGM scrubber to remove RGM before the

634 measurement of Hg(0). This inline scrubber utilizes deposition on uncoated quartz wool and the results of

635 Ambrose et al. (2013) imply that while uncoated quartz captures RGM efficiently in the presence of $O_3$,

636 quartz with a KCl coating promotes efficient reduction to Hg(0).

637  It is also reasonable to question the extent to which the Tekran speciation systems operated at

638 RAMIX reflect the performance of these systems when normally operated under  protocols. As noted

639 above, the operation of the RAMIX manifold and the Tekran speciation systems has been questioned by

640 Prestbo (2014). In our view the two most significant issues are the performance of the two 2537 mercury

641 analyzers associated with each speciation system and the reduced sampling rate. The performance of the

642 two 2537 units is detailed in Gustin et al. (2013) and, as they noted, there was a significant response in each

643 instrument. Examination of Fig SI 6 of Gustin et al. (2013) shows the relative responses of the two

644 instruments and, using concentrations up to 25 ng $m^{-3}$ i.e. manifold spikes, they list a regression of 0.72

645 [Hg(0)] + 0.08 whereas for the non-spike data they obtain 0.62[Hg(0)] + 0.25. Their Table SI 5 lists the

646 regression including spikes as 0.7 (±0.01) + 0.2, with all concentrations expressed in ng $m^{-3}$.  When

647 considering the use of these analyzers to monitor oxidized mercury the important factor to consider is the

648 loading on the gold cartridge. Table SI 3 lists the mean RGM concentrations from manifold sampling as 52

649 pg $m^{-3}$ for SPEC1 and 56 pg $m^{-3}$ for SPEC2. For a 1 hour sample at 4 L $min^{-1}$ this corresponds to a cartridge

650 loading of 13 pg. This is similar to the cartridge loading for sampling a concentration of 0.6 ng $m^{-3}$ at 4 l

651 $min^{-1}$ for 5 minutes. If we examine Fig SI 6 of Gustin et al. (2013) we see that the regression analyses are

652 based on higher concentrations than 0.6 ng $m^{-3}$, i.e. higher cartridge loadings. At concentrations of 0.6 ng

653 $m^{-3}$ the ratio of SPEC2:SPEC1 obtained from these regressions would be 1.05, 0.85 and 1.06 depending on

654 which regression formula is used.  We should note that based on Table SI 6 the median RGM

655 concentrations in manifold sampling were 41 and 46 pg $m^{-3}$. The RGM concentrations for free standing

656 sampling were even lower with means of 26 and 19  pg $m^{-3}$ and medians of 23 and 14 pg $m^{-3}$ for SPEC1

657 and SPEC2 respectively.  For concentrations below 40 pg $m^{-3}$ the cartridge loading drops below 10 pg and

658 in addition, the Tekran 2537 integration routine becomes significant. Swartzendruber et al. (2009) reported

659 issues with the standard integration routine and note that below cartridge loadings of 10 pg the internal

660 integration routine produces a low bias in the Hg(0) concentration. They recommend downloading the raw

661 data, i.e. PMT output and integrating offline.  This issue has recently been discussed by Slemr et al. (2016)

662 in a reanalysis of data from the CARIBIC program. This compounds the problem of correcting the bias





between SPEC1 and SPEC2.  Because the speciation instruments were sampling at 4 L /min rather than the
recommended 10 L/min a large number the measurements made by the speciation systems are based on
uncorrected cartridge loadings of less than 10 pg m$^{-3}$. Based on the above we caution against drawing
significant conclusions based on differences between SPEC1 and the corrected SPEC2. These differences
are the basis of the conclusions of Gustin et al. (2013) that "On the basis of collective assessment of the
data, we hypothesize that reactions forming RM (reactive mercury)  were occurring in the manifold"
(Gustin et al. (2013) abstract). Later they state "The same two denuders, coated by the same operator, were
used from Sept 2 to 13, and these were switched between instruments on September 9. Prior to switching
the slope for the equation comparing GOM as measured by Spec 1 versus Spec 2 adjusted was 1.7
($r2=0.57$, $p<0.5$, $n=76$) after switching this was 1.2 ($r2=0.62$, $p<0.05$, $n=42$). This indicates that although
there may have been some systematic bias between denuders SPEC 2 adjusted consistently measured more
GOM than SPEC 1. We hypothesize that this trend is due to production of RM in the manifold (discussed
later)." If reactions in the manifold were producing RM then this production would surely have resulted in
the DOHGS measuring artificially high, i.e. higher than ambient, concentrations of oxidized mercury.
However, the paper by Ambrose et al. (2013) (written by a subset of the authors of Gustin et al.(2013))
makes no mention of manifold production of oxidized mercury. In fact Ambose et al. (2013) state, in the
supplementary information to their paper, "The same two denuders, prepared by the same operator, were
used in the Tekran® Hg speciation systems from 2 to 13 September. The denuders were switched between
Spec. 1 and Spec. 2 on 9 September. From 2 to 9 September, the Spec. 1-GOM/Spec. 2-GOM linear
regression slope was 1.7 ($r2 = 0.57$; $p < 0.05$; $n = 76$); from 9 to 13 September the Spec. 1-GOM/Spec. 2-
GOM slope was 1.2 ($r2 = 0.62$; $p < 0.05$; $n = 42$). These results suggest that the precisions of the GOM
measurements made with Spec. 1 and Spec. 2 were limited largely by inconsistent denuder performance."

The oxidized mercury concentrations presented by Ambrose et al. (2013) for the RAMIX

measurements suggests a well-defined diurnal profile that peaks at night. It is important to note that the
error bars on this profile (Figure 3 of Ambose et al.) are one standard error rather than one standard
deviation. The standard deviations, which actually give an indication of the range of concentrations
measured show much larger errors indicating significant day to day variation in these profiles.
Nevertheless, the measurements show much larger oxidized mercury concentrations than the speciation
systems and the very limited number of 2P-LIF measurements.  As we note below, there is no known or
hypothesized chemistry that can reasonably explain the large RGM concentrations seen by the DOHGS
instrument. Both Gustin et al. (2013) and Ambrose et al. (2013) draw some conclusions about the chemistry
of mercury that have significant implications for atmospheric cycling. Gustin et al. suggest in their abstract
that "On the basis of collective assessment of the data, we hypothesize that reactions forming RM were
occurring in the manifold." Later in a section on "Implications" they conclude "The lack of recovery of the
HgBr$_2$ spike suggests manifold reactions were removing this form before reaching the instruments." The
residence time in the RAMIX manifold was on the order of 1s depending on sampling point and there is no
known chemistry that can account for oxidation of Hg(0) or reduction of RGM on this timescale. We would



suggest that the most reasonable explanation of the discrepancies between the various RAMIX
measurements includes both instrumental artifacts and an incomplete characterization of the RAMIX
manifold. If fast gas-phase chemistry is producing or removing RGM in the RAMIX manifold the same
chemistry must be operative in the atmosphere as a whole and this requires that we completely revise our
current understanding of mercury chemistry. The discrepancies between the DOHGS and speciation
systems are further indication that artifacts are associated with KCl denuder sampling under ambient
conditions but we would suggest that RAMIX does not constitute an independent verification of the
DOHGS performance and that the 2P-LIF measurements raise questions about the DOHGS measurements.
Ambrose et al. (2013) also suggest that the observations of very high RGM concentrations indicate
multiple forms of RGM and that the concentrations can be explained by oxidation of Hg(0), with $O_3$ and
$NO_3$ being the likely nighttime oxidants.  We have discussed these reactions in detail previously (Hynes et
al., 2009) and concluded that they cannot play any role in homogeneous gas phase oxidation of Hg(0).
Ambrose et al. (2013) cite recent work on this reaction by Rutter et al. (2012) stating that "On the basis of
thermodynamic data for proposed reaction mechanisms, purely gas-phase Hg(0) oxidation by either $O_3$ or
$NO_3$ is expected to be negligibly slow under atmospheric conditions; however, in the case of $O_3$-initiated
Hg(0) oxidation, the results of laboratory kinetics studies unanimously suggest the existence of a gas-phase
mechanism for which the kinetics can be treated as second-order." We would suggest that a careful reading
of the cited work by Rutter et al. (2013) demonstrates the opposite conclusion. We provide additional
discussion of these issues in the SI and again conclude that $O_3$ and $NO_3$ can play no role in the
homogeneous gas phase oxidation of Hg(0).

**5.0 Future Mercury Intercomparisons:**
The discrepancies that are discussed above suggest a need for a careful independent evaluation of
mercury measurement techniques. The approaches used during the evaluation of instrumentation for the
NASA Global Tropospheric Experiment (GTE) and the Gas-Phase Sulfur Intercomparison Experiment
(GASIE) evaluation offer good models for such an evaluation. The Chemical Instrument and Testing
Experiments (CITE 1-3) (Beck et al., 1987; Hoell et al., 1990; Hoell et al., 1993) were a major component
of GTE establishing the validity of the airborne measurement techniques used in the campaign. The GASIE
experiment (Luther and Stetcher, 1997; Stetcher et al., 1997) was a ground based intercomparison of $SO_2$
measurement techniques that might be particularly relevant to issues associated with mercury measurement.
In particular, GASIE was a rigorously blind intercomparison that was overseen by an independent panel
consisting of three atmospheric scientists none of whom were involved in $SO_2$ research. We would suggest
that a future mercury intercomparison should be blind with independent oversight. Based on the RAMIX
results it should consist of a period of direct ambient sampling and then manifold sampling in both reactive
and unreactive configurations. For example an unreactive configuration would consist of Hg(0) and
oxidized mercury in an $N_2$ diluent eliminating any possibility of manifold reactions and offering the
possibility of obtaining a manifold blank response. Such a configuration would allow the use of both



denuder and pyrolysis measurements since it is reasonable to conclude, based on the current body of
experimental evidence, that denuder artifacts are associated with ambient sampling with water vapor and
ozone as the most likely culprits. A reactive configuration would be similar to the RAMIX manifold
configuration with atmospheric sampling into the manifold and periodic addition of Hg(0) and oxidized
mercury over  their ambient concentrations. The combination of the three sampling configurations should
enable instrumental artifacts to be distinguished from reactive chemistry in either the manifold itself or, for
example, on the KCl denuder.
**6.0 Conclusions**
We deployed a 2P-LIF instrument for the measurement of Hg(0) and RGM during the RAMIX campaign.
The Hg(0) measurements agreed reasonably well with instruments using gold amalgamation sampling
coupled with CVAFS analysis of Hg(0). Measurements agreed to 10-25% on the short term variability in
Hg(0) concentrations based on a 5 minute temporal resolution. Our results also suggest that the operation of
the RAMIX manifold and spiking systems were not as well characterized as Finley et al. (2013) suggest.
We find that the calculated concentration spikes consistently overestimated the amount of Hg(0) introduced
into the RAMIX manifold by as much as 30%. This suggests a systematic error in concentration
calculations rather than random uncertainties that should not produce a high or low bias.
We made  measurements of TGM, and hence TOM by difference, by using pyrolysis to convert
TOM to Hg(0) and switching between pyrolyzed and ambient samples. The short term variation in ambient
Hg(0) concentrations is a significant limitation on detection sensitivity and suggests that a two channel
detection system, monitoring both the pyrolyzed and ambient channels simultaneously is necessary for
ambient TOM measurements. Our TOM measurements were normally consistent, within the large
uncertainty, with KCl denuder measurements obtained with two Tekran Speciation Systems and with our
own manual KCl denuder measurements. The ability of the pyrolysis system to measure higher RGM
concentrations was demonstrated during one of the manifold HgBr$_2$ spikes but the results did not agree with
those reported by the UW DOHGS system. We would suggest that it is not possible to reconcile the
different measurement approaches to TOM. While there is other evidence that KCl denuders may
experience artifacts in the presence of water vapor and ozone the reported discrepancies cannot explain the
very large differences reported by the DOHGS and Tekran speciation systems. Similarly, the differences
between the DOHGS and 2P-LIF pyrolysis measurements suggest that one or both of the instruments were
not making reliable, quantitative measurements of RGM. We suggest that both instrumental artifacts, an
incomplete characterization of the sampling manifold, and limitations in the measurement protocols make
significant contributions to the discrepancies between the different instruments and that it would be rash to
draw significant implications for the atmospheric cycling of mercury based on the RAMIX results.  This is
particularly true of the RGM results. If one were to conclude that the discrepancies between the DOHGS
and speciation systems sampling ambient oxidized mercury are accurate and reflect a bias that can be
extrapolated to global measurements then it means that atmospheric RGM concentrations are much higher
than previously thought and that we have little understanding of the atmospheric cycling of mercury.  What



is not in dispute is the urgent need to resolve the discrepancies between the various measurement
techniques. The RAMIX campaign provided a valuable guide for the format of any future mercury
intercomparison. It clearly demonstrated the need to deploy high accuracy calibration sources of Hg(0) and
oxidized mercury, the need for multiple independent methods to measure elemental and oxidized mercury
and to clearly characterize and understand the differences reported by instruments that are currently being
deployed for measurements.

**Acknowledgements**
This work was supported by NSF Grant # AGS-1101965, a National Science Foundation Major
Instrumental Grant (#MRI-0821174) and by the Electric Power Research Institute. We thank Mae Gustin
and her research group and Dr Robert Novak for their hospitality, assistance and use of laboratory facilities
during the RAMIX intercomparison. We thank Mae Gustin and Dan Jaffe for the use of their RAMIX data
for comparison with our results. We thank Eric Prestbo for helpful comments on the manuscript.

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

Table 1: RAMIX Manual KCl Denuder Sampling

| Date | Sample time | mid point | sample | blank | | time | spec1 | | spec2 (uncorr | |
|---|---|---|---|---|---|---|---|---|---|---|
| | hours | hour | pg m$^{-3}$ | pg m$^{-3}$ | | | GOM | PBM | GOM | PBM |
| | | | | | | | pg m-3 | pg m-3 | pg m$^{-3}$ | pg m$^{-3}$ |
| 9/6 | 1.5 | 15 | 127.9* | 2.27 | | 13:00 | 200.7 | 51.8 | 205.1 | 4.3 |
| | | | | | | 15:00 | 65.7 | 32.0 | 84.9 | 6.0 |
| | | | | | | | | | | |
| 9/7 | 2 | 16 | 112.9* | 0 | | 14:00 | 39.8 | 136.4 | 94.3 | 2.5 |
| | | | 21.2 | | | 16:00 | 48.5 | 177.3 | 68.9 | 1.5 |
| | | | 285.8* | | | 18:00 | 28.1 | 182.2 | 37.4 | 3.3 |
| | | | 30.6 | | | | | | | |
| | | | | | | | | | | |
| 9/10 | 3 | 15.3 | 74.3 | 1995 | | 14:00 | 26.7 | 10.5 | 27.4 | 4.2 |
| | | | 44.2 | | | 16:00 | 24.1 | 18.3 | 23.7 | 2.3 |
| | | | | | | | | | | |
| 9/13 | 4 | 15 | 12.8 | 8.2 | | 13:00 | 0.7 | 16.9 | 0.5 | 16.6 |
| | | | 13.56 | | | 17:00 | 37.6 | 16.1 | 25.2 | 2.7 |
| | | | | | | | | | | |
| 9/14 | 4.5 | 14 | 39* | 3.3 | | 12:00 | 34.9 | 12.0 | 23.9 | 5.5 |
| | | | 17.3 | | | 14:00 | 57. | 18.4 | 26.3 | 38.6 |
| | | | | | | 16:00 | 42.0 | 17.4 | 26.3 | 4.0 |
| | | | | | | | | | | |
| 9/15 | 4.5 | 15 | 15.24 | 1.53 | | 13:00 | 113.9 | 39.1 | 27.6 | 3.9 |
| | | | 20.4 | 4.87 | | 15:00 | 80.6 | 22.2 | 17.7 | 3.9 |
| | | | | | | 17:00 | 110.8 | 24.1 | 8.6 | 8.1 |
| | | | | | | | | | | |





| 9/16 | 2.75 | 16 | 148* | 5 | | 8:00 | 19.7 | 4.7 | 14.8 | 5.4 |
|------|------|-----|------|---|---|-------|------|------|------|-----|
| | | | 42 | 6 | | 9:00 | | | | |
| | | | 26 | 5 | | 10:00 | 28.7 | 13.3 | 19.9 | 4.8 |
| | | | 47 | 4 | | | | | | |

•    * evidence from TDP's for presence of PBM
•    Measurements for UNR Speciation system made at similar times. The Spec 2 measurements are
uncorrected values.
















**Figures.**




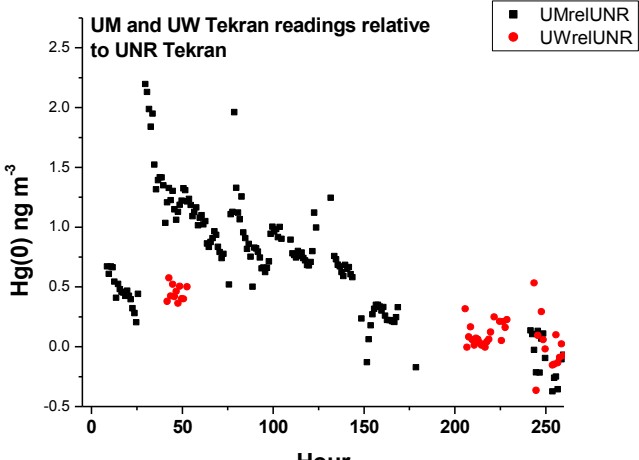


Figure 1. Comparison of Hg(0) readings from the UM, UW and UNR Tekrans over the first 260 hours of
UM measurements. The absolute concentration difference relative to the UNR instrument is shown in black
for the UM Tekran and in red for the DOHGS (UW) Tekran.

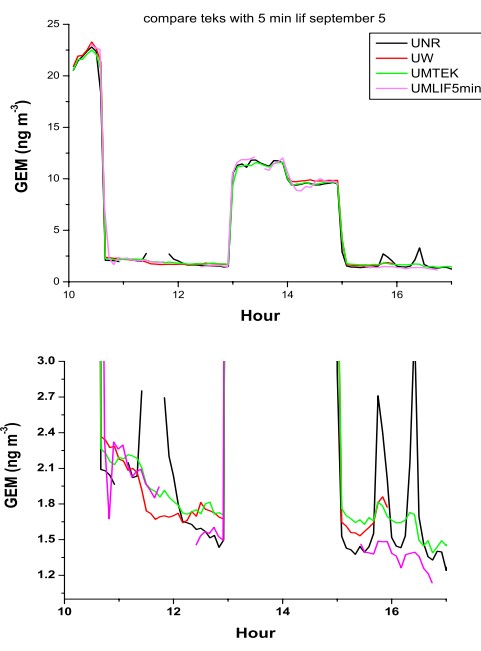






Figure 2: a). A seven hour sequence of GEM measurements from September 5[th] that included two manifold
spikes. Shown are the sequence of GEM measurements from the UNR, UW and UM Tekrans together with
the 5 minute averages of the 2P-LIF signal.  b) An expanded concentration scale focusing on ambient
measurements.


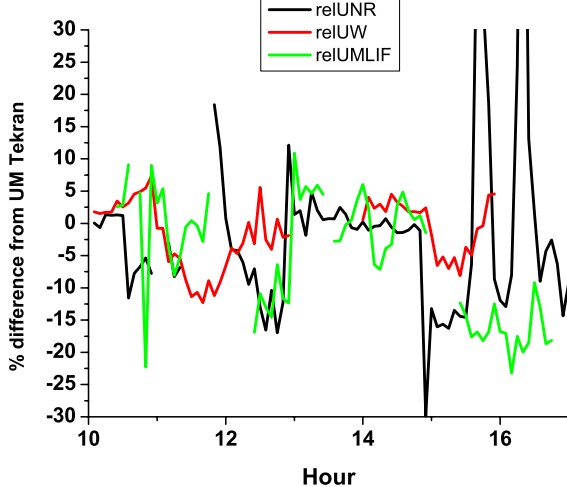



Figure 3: Seven hour measurement period from September 5[th]. The % difference of the UNR (black line)
and UW (red line) Tekrans and the UM 2P-LIF (blue line) measurements relative to the UM Tekran is
shown.














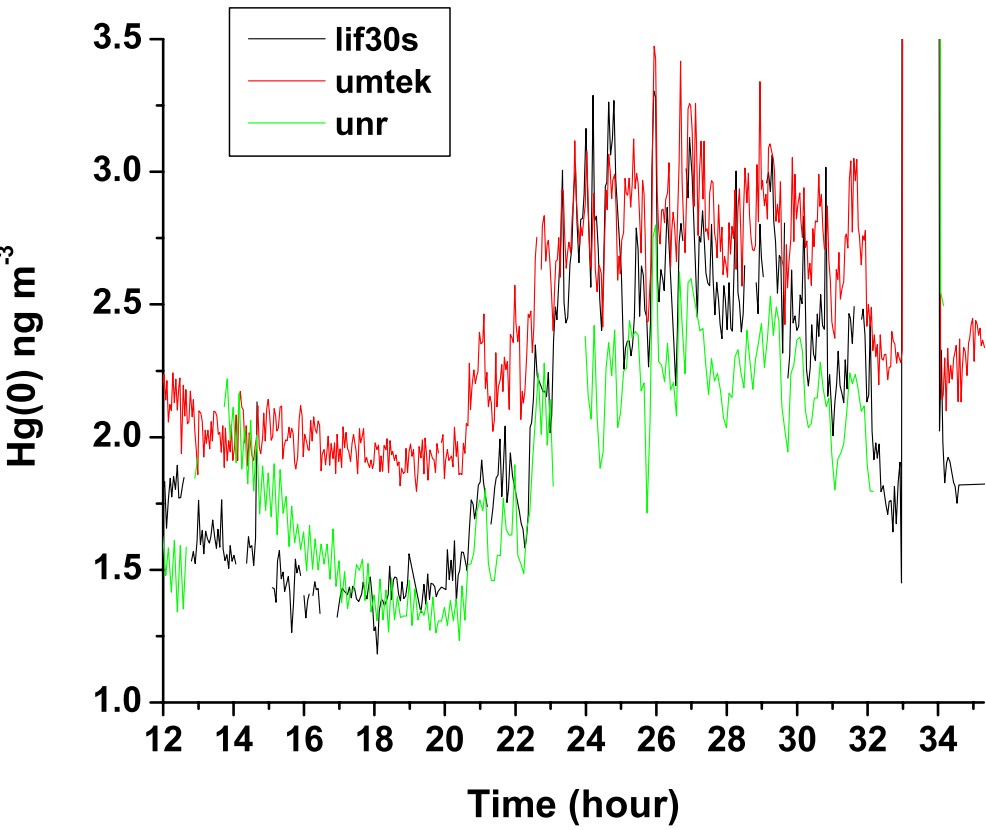

Fig 4: 22 hour sampling period from September 1[st] and 2[nd]. Comparison of the UM (red line) and UNR
(green line) Tekrans with the UM 2P-LIF (black line) concentrations. The concentrations for each
instrument are scaled to force agreement during the second manifold spike at hour 33. This is the data from
SI Fig. 3 with the concentration scale expanded to shown only ambient data.





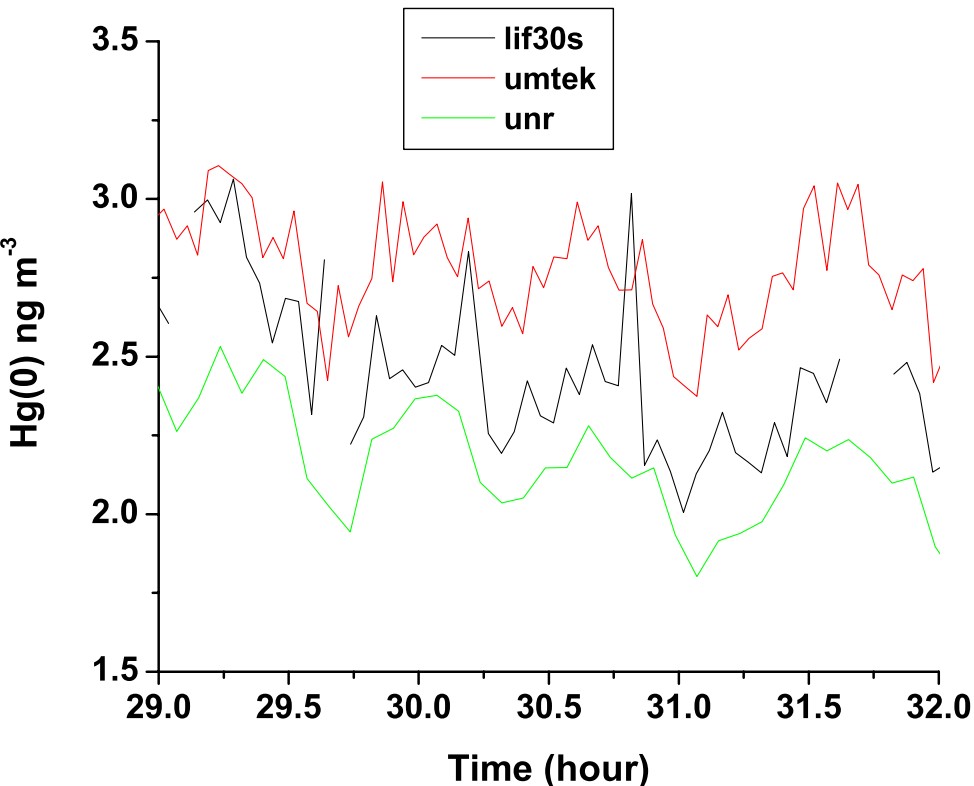

Fig 5: A section of the 22 hour sampling period from September 1st and 2nd. Comparison of the UM (red
line) and UNR (green line) Tekrans with the UM 2P-LIF (black line) concentrations. The concentrations
for each instrument are scaled to force agreement during the second manifold spike at hour 33. This is the
data from SI Fig. 3 with the concentration scale expanded to shown only ambient data between hours 29
and 32.





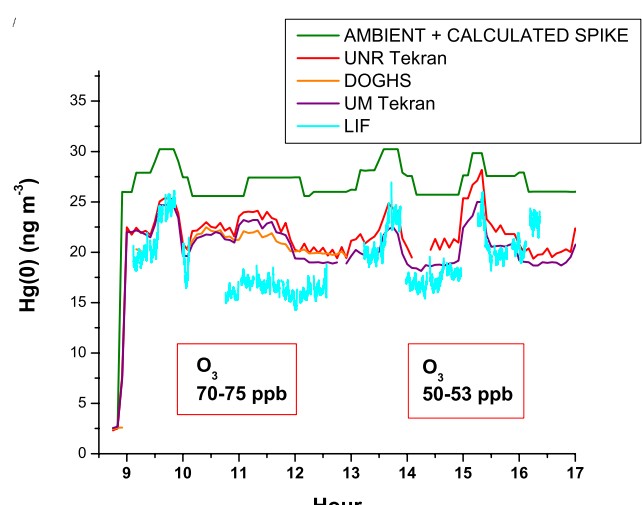



Figure 6: September 7$^{th}$ an ozone interference test. A comparison of the UM, UW and UNR Tekrans and the UM-2P-LIF measurements. The "expected" concentration calculated from the ambient Hg(0) concentration prior to the spike plus the calculated spike concentration is also shown.





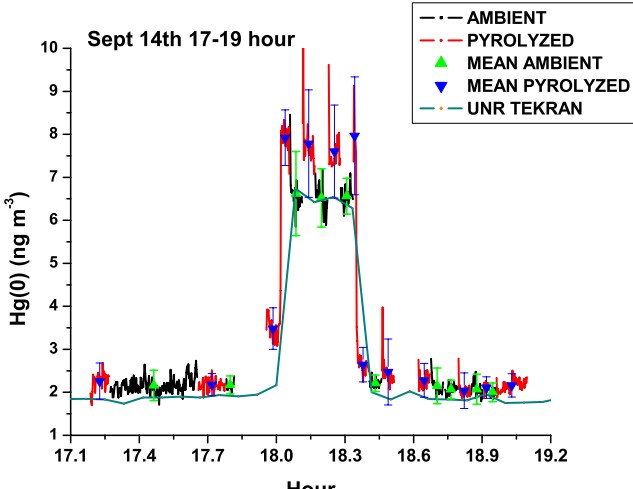



Fig 7: September 14 measurements  hours 17-19 (5-7pm). The background subtracted 2P-LIF signals from
the ambient (black) and pyrolyzed sampling lines (red) are shown. The gaps correspond to times when the
laser was blocked to check power and background. The means and 1 standard deviation of each sample are
shown. The absolute Hg(0) concentrations are obtained by scaling the ambient Hg(0) signal to the absolute
Hg(0) concentration reported by the UNR Tekran during the Hg(0) manifold spike.




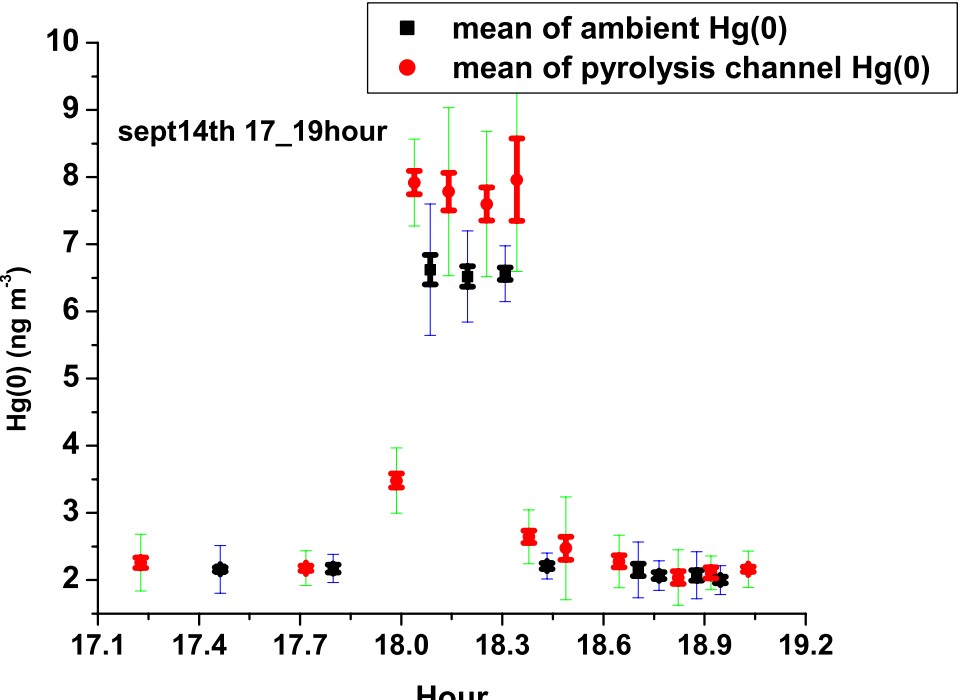



Fig. 8: September 14 measurements hour 17-19. The means of the ambient channel (black) and pyrolyzed
channel (red) are shown. The error bars show both 2 standard errors (thicker line) and 2 standard
deviations.





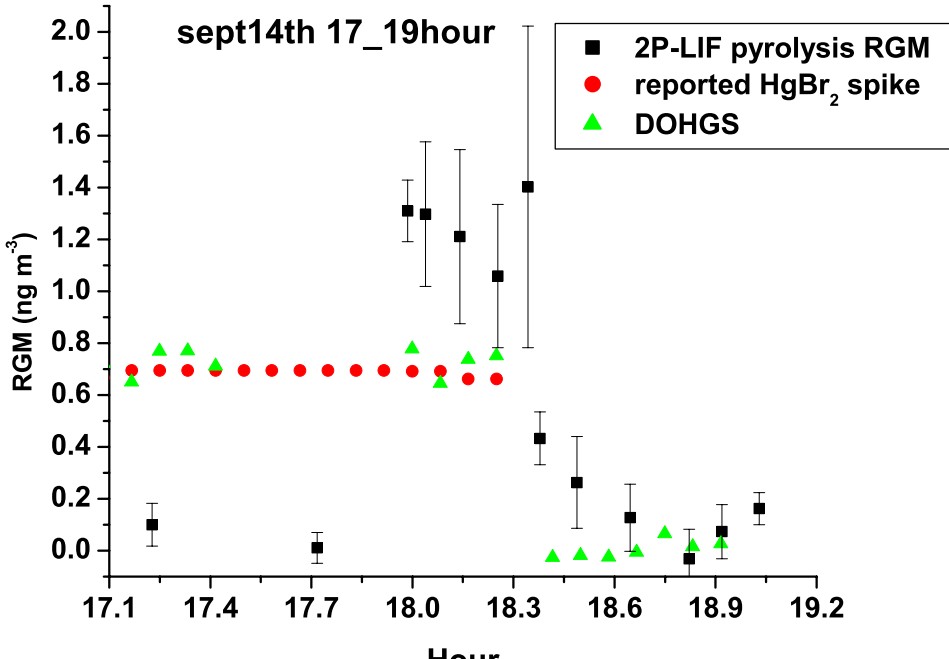

Fig 9: TOM concentrations calculated from the difference between the pyrolyzed and ambient sample
concentrations together with 2SE in the TOM concentrations. The reported $HgBr_2$ spike concentrations and
DOHGS measurements are also shown.






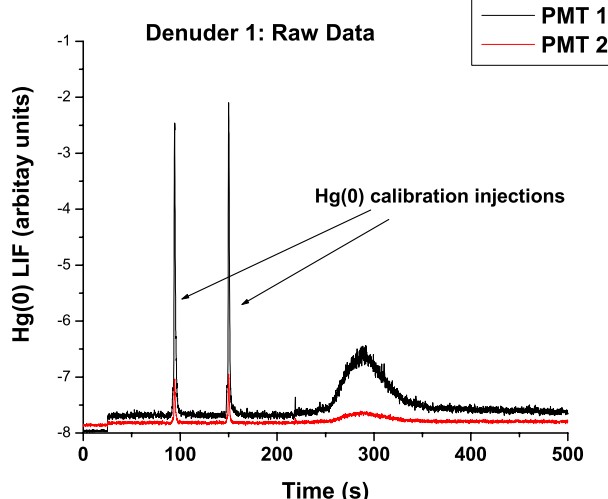

Fig. 10: September 16[th] KCl manual denuder measurements. The raw data for the temporal decomposition
profiles (TDP) for the denuder D1 is shown.





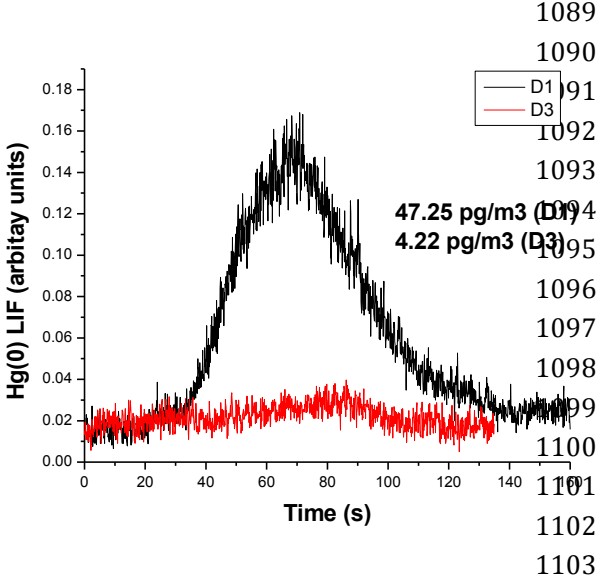

Fig. 11 September
16[th] KCl manual
denuder measurements. The calibrated temporal decomposition profiles (TDP) for the tandem denuder pair,
D1 and D3 are shown.






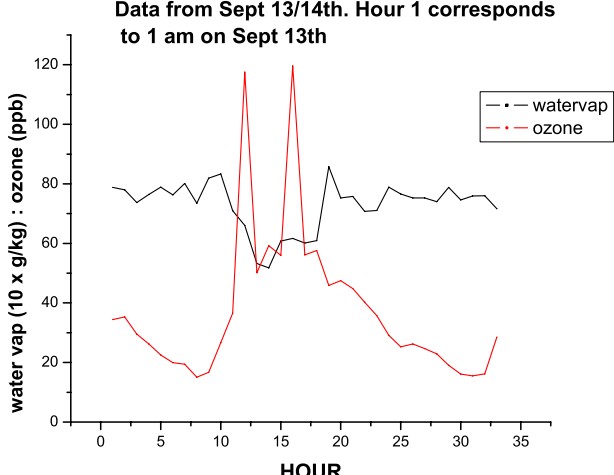

Fig. 12: The ozone concentration and absolute humidity for a 35 hour sampling period on September 13[th]
and 14[th] that included two ozone spikes and only sampled ambient TOM.





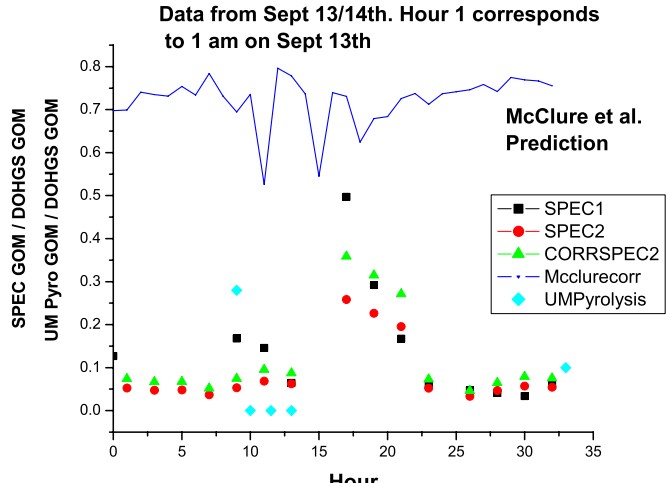

Fig. 13. Expected denuder recovery based on the formula determined by McClure et al. which varies
between a typical value of ~70% dropping to ~50% during the ozone spikes. The figure also shows the
reported recoveries i.e. the ratio of RGM as measured by either the UNR speciation systems or the 2P-LIF
system devided by the value reported by the DOHGS system.