# Peer review of "In-situ and Denuder Based Measurements of Elemental and Reactive 1 2 Gaseous Mercury with Analysis by Laser-Induced Fluorescence. Results from the Reno Atmospheric Mercury Intercomparison Experiment. 3 4 Anthony J. Hynes\*, Stephanie Everhart, Dieter Bauer, James Remeika, and Cheryl Tatum"

_Atmospheric Chemistry and Physics, 2016_

## Referee Comment (RC1) · Anonymous Referee #1 · 15 Aug 2016

This manuscript summarizes the results obtained with a two-photon technique applied to measuring atmospheric mercury. It put a new perspective on the results of the RAMIX inter-comparison which in my opinion is more accurate than two previously published papers.

P. 4, line 124 – Your inlet was 25 feet long and made of Teflon (type unknown). Teflon is OK for passing elemental mercury, but it has memory problems when going from high to low mixing ratios. Did you determine the passing efficiency of the tubing that

was used and check for memory (passing zero air through it after sampling air)? This memory problem is not obvious unless it is carefully checked. It may have influenced some of your results. RGM will also stick to that length of Teflon tubing. This may also have influenced your results (TGM).

P. 4, Section 2.3 – Measuring TOM by difference is tricky to accomplish accurately, especially in ambient air. What was your estimated LOD for TOM using this method? How was the uncertainty determined?

P. 6, lines 184 . . . - These syringe injections can be difficult to duplicate with high precision. Did this add additional uncertainty to your data?

P. 7, lines 230-233 – I agree that these changes are highly unlikely to be caused by chemical reactions in the manifold. This was a dark environment presumably without photochemical processes occurring.

P. 8, line 258 – I agree that the calibration of the elemental mercury spikes in the manifold must of have much higher uncertainty than stated by the operators.

P. 8, lines 273-274 – This is hard to rationalize and it points to the need for a new blind inter-comparison done with third party observers as suggested in this manuscript. The NSF and NASA inter-comparisons are an excellent example of how these should be conducted.

P. 16, lines 563 . . . - This all points to the need for better calibration of all instruments and a more carefully conducted inter-comparison. The LIF system is capable of finding problems not apparent with the two Tekrans.

In summary, this manuscript emphasizes the need for a more carefully conducted inter-comparison for atmospheric mercury.

---

## Author Comment (AC1) · 15 Sep 2016

Reply to Referee #1.

We appreciate the comments of the referee and we include our responses below the comments.

P. 4, line 124 – Your inlet was 25 feet long and made of Teflon (type unknown). Teflon is OK for passing elemental mercury, but it has memory problems when going from

high to low mixing ratios. Did you determine the passing efficiency of the tubing that was used and check for memory (passing zero air through it after sampling air)? This memory problem is not obvious unless it is carefully checked. It may have influenced some of your results. RGM will also stick to that length of Teflon tubing. This may also have influenced your results (TGM).

Reply: We believe that the data from Fig. 2 (September 5th) best address the issue of passing efficiency and memory effect. The three independently operated Tekrans and the LIF system show excellent agreement switching from ~23 ng m3 to ambient, and then from ~ 10 ng m-3 to ambient. All three instruments had very different sampling lines, with the lines to the UW DOAGHS and the UNR Tekran being much shorter than the ~25 Ft sampling line to our instrument. We noted, and Fig 2 shows clearly, that at the completion of the second spike all the instruments drop to ambient but the UNR instrument sees two Hg(0) "pulses" that show up with greatly reduced amplitudes in the UW and UM Tekran signals and also in the 2P-LIF signal. These occur ~49 and 88 minutes after the spike and we do not think that these are associated with memory effects.

It is certainly possible that reduction of RGM on the walls could influence the results. We noted in section 3.3.1 "It is not clear if the deviations that are observed, particularly the large deviations seen by the UNR Tekran after the second spike are related to presence of elevated levels of HgBr2, or other issues related to manifold operation." We also noted in section 3.4 that "As we have noted previously (Bauer et al., 2014), condensation in our sampling lines can produce artifacts in Hg(0) concentration measurements. Because of the low humidity in Reno it was not necessary to use any type of cold trap during ambient measurements but we did use a trap during manifold spikes of H2O so our measurements do not address this as a potential interference." When condensation occurs in the sampling lines we see increases in Hg(0) that are presumably related to reduction of RGM deposited on the walls of the sampling lines. We have only seen these effects when we have condensation but these humidity effects merit

further investigation.

P. 4, Section 2.3 – Measuring TOM by difference is tricky to accomplish accurately, especially in ambient air. What was your estimated LOD for TOM using this method? How was the uncertainty determined?

Reply: We would certainly agree that this approach is tricky! As we noted in section 3.5 "TOM was calculated from the difference in the TGM and Hg(0) concentrations and in this sampling configuration the limit of detection for TOM depends on the short term variability in ambient Hg(0) which is significant and shows a diurnal variation." and "We calculated the means of the pyrolysis and ambient channel concentrations, and the difference which gives the TOM concentration. We also calculated the standard deviations and standard errors (SE) and used these errors to calculate in quadrature the 2SE uncertainty in the derived TOM concentration. However, as discussed below, the errors in the means do not appear to capture the full variability in Hg(0), particularly at shorter sampling times." The 2SE uncertainties were typically on the order of 50-100 pg m-3 but we again emphasize that these seem to underestimate the real uncertainty associated with Hg(0) variability. We concluded in section 3.6 "Our results suggest that the use of single detection channel with switching between ambient and pyrolyzed samples is not adequate to resolve the small concentration differences that are necessary to be able to monitor ambient TOM. It is necessary to set up two detection systems, one continuously monitoring ambient Hg(0) and the other continuously monitoring a pyrolyzed sample stream giving TGM, to get the precision necessary to monitor ambient TOM."

P. 6, lines 184 ...  - These syringe injections can be difficult to duplicate with high precision. Did this add additional uncertainty to your data?

Reply: Figure 10 shows an example of data including two syringe injections. The difference in the area of these calibration injections is ∼1.5%. The differences in duplicate samples from denuders that are not affected by particulate sampling ( i.e. 3 of the

September 16th samples in Table 1) are significantly larger than this so we do not believe the calibration method adds significant uncertainty to this data.

We concur with the remaining comments:

P. 7, lines 230-233 – I agree that these changes are highly unlikely to be caused by chemical reactions in the manifold. This was a dark environment presumably without photochemical processes occurring. P. 8, line 258 – I agree that the calibration of the elemental mercury spikes in the manifold must of have much higher uncertainty than stated by the operators. P. 8, lines 273-274 – This is hard to rationalize and it points to the need for a new blind inter-comparison done with third party observers as suggested in this manuscript. The NSF and NASA inter-comparisons are an excellent example of how these should be conducted. P. 16, lines 563 . . . - This all points to the need for better calibration of all instruments and a more carefully conducted inter-comparison. The LIF system is capable of finding problems not apparent with the two Tekrans. In summary, this manuscript emphasizes the need for a more carefully conducted inter-comparison for atmospheric mercury.

---

## Referee Comment (RC2) · Anonymous Referee #2 · 21 Sep 2016

Review of Manuscript

In-situ and Denuder Based Measurements of Elemental and Reactive Gaseous Mercury with Analysis by Laser-Induced Fluorescence. Results from the Reno Atmospheric Mercury Intercomparison Experiment (acp-2016-446)

This manuscript describes the work of Hynes et al., to quantify elemental gaseous mercury (Hg0) and total gaseous mercury (TGM) concentrations using a sequential

two photon laser-induced fluorescence (2P-LIF) instrument off a manifold as part of the RAMIX method inter-comparison study conducted in Reno, NV. As the authors point out, there is currently a debate in the literature concerning the efficacy of various ambient mercury measurement methods under different conditions of ambient relevance. As such the RAMIX study endeavored to provide a platform for a definitive methods comparison. Unfortunately, based on Hynes et al work (and references within) the RAMIX study and the manifold delivery system designed and implemented for this study fell short of this goal in several important aspects that limits the utility of the study's findings. As a result, I believe this paper is an important contribution to the state-of-science. I also have some technical concerns/questions with the implementation of some of the experiments described in the paper enumerated in the comments below, therefore I feel this manuscript will require substantive revision before it is acceptable for publication in ACP.

See enumerated general and specific comments in attachment.

Please also note the supplement to this comment:
http://www.atmos-chem-phys-discuss.net/acp-2016-446/acp-2016-446-RC2-supplement.pdf

**Supplement:**

Review of Manuscript

In-situ and Denuder Based Measurements of Elemental and Reactive Gaseous Mercury with Analysis by Laser-Induced Fluorescence. Results from the Reno Atmospheric Mercury Intercomparison Experiment (acp-2016-446)

This manuscript describes the work of Hynes et al., to quantify elemental gaseous mercury ($Hg^0$) and total gaseous mercury (TGM) concentrations using a sequential two photon laser-induced fluorescence (2P-LIF) instrument off a manifold as part of the RAMIX method inter-comparison study conducted in Reno, NV. As the authors point out, there is currently a debate in the literature concerning the efficacy of various ambient mercury measurement methods under different conditions of ambient relevance. As such the RAMIX study endeavored to provide a platform for a definitive methods comparison. Unfortunately, based on Hynes et al work (and references within) the RAMIX study and the manifold delivery system designed and implemented for this study fell short of this goal in several important aspects that limits the utility of the study's findings. As a result, I believe this paper is an important contribution to the state-of-science. I also have some technical concerns/questions with the implementation of some of the experiments described in the paper enumerated in the comments below, therefore I feel this manuscript will require substantive revision before it is acceptable for publication in ACP.

General Comments:

(1) In several areas of the paper (e.g., lines 63-64; 144-147) the authors incorrectly include particulate bound mercury (PBM) into their definition of total gaseous mercury (TGM). In the literature TGM is generally used to describe $Hg^0$ + divalent reactive gaseous mercury (RGM). In fact most ambient instruments that quantify TGM (e.g., Tekran Instruments Corporation Model 2537) use an integrated Teflon filter to exclude all particulate matter, and most monitoring networks include an integrated Teflon filter at the inlet of their sampling line to minimize gas/particle interactions in the sampling line that has been shown to create problems in reliably quantifying ambient gaseous mercury species (see below). If the system is pyrolyzing all ambient mercury species to $Hg^0$ for detection, then perhaps coin a new operationally defined term such as total atmospheric mercury (TAM) to avoid confusion.

(2) The authors need to provide additional details on their sampling configuration and calibration procedures (QA/QC) in the methods section to provide readers and reviewers the basic information necessary to inform and judge the implementation of the their 2P-LIF system.

(3) Though not a problem with this paper per se, the design and implementation of the RAMIX study manifold system described by Finley et al. (2013) for this work is problematic and the results described in this paper must be viewed through this lens. The overarching issue was the decision to allow ambient particulate matter ($PM_1$) into the manifold. When the stated objective was to evaluate the efficacy of different measurement methodologies to quantify $Hg^0$ and RGM the logical first step would have been to evaluate these gaseous species first without the complication of spiked gases interacting with ambient particles in the manifold. The curious choice of constructing the manifold using highly porous PFA Teflon tubing also creates multiple potential problems (i) absorption/desorption of spiked gas concentrations following diffusion gradients, (ii) non-conducting material combined with high (187 LPM) flow rate and low relative humidity could lead to electrostatic collection of PM on the internal manifold walls providing additional surfaces for gas phase adsorption, and (iii) poor conduction of heat applied to the external surface of the tubing at such a high manifold flow rate (Finley et al. reported using eight thermocouples to measure external temperature down the length of the manifold but did not report any measuring internal surface or air temperature). Using a blower to maintain flow through the manifold also added a reported 15% uncertainty in the spike concentrations (Finley et al., 2013). All these issues lead to relatively low reported average transmission efficiencies for $Hg^0$ (92%), $HgBr_2$ (76%), and $O_3$ (93%) even under controlled laboratory conditions (Finley et al., 2013). The range for $Hg^0$ transmission in the manifold was reported to be 71-101% by Prestbo (2014). If the spiked mass of target gases into the manifold were not conserved through the system, then definitive evaluation of the analytical instruments sampling from the manifold were compromised. I agree with the authors that 15% probably underestimates the overall uncertainty for $Hg^0$ (Lines 205-207) and that the system should be characterized as "a semi-quantitative delivery system" (Lines 210-211). If the manifold cannot quantitatively and reproducibly transmit a relatively inert gas like $Hg^0$ there is very little chance of reproducibly transmitting $HgBr_2$.

(4) This reviewer agrees with the authors that previous RAMIX study papers invoke and discuss mercury oxidation and reduction chemistry that is not supported in either the theoretical or applied literature. Controlling the physical adsorption of gaseous species onto manifold surfaces and aerosols is the logical first direction for which to find answers.

Specific Comments:

(1) Introduction:
   a. Lines 49-50: Sprovieri et al., 2016 Atmos. Chem. Phys. Discuss., doi:10.5194/acp-2016-466 is a more contemporary discussion of global background $Hg^0$ concentrations in the Northern and Southern hemispheres from the global GMOS network.
   b. Lines 63-64: PBM is not part of TGM.
(2) Experimental:

a. As indicated above, the authors need to supplement the QA/QC information.

    i. Lines 122-126: The authors discuss calibrating the 2P-LIF system using a Tekran 2537B as a secondary transfer standard, and that unlike their previous field work the unit was turned off for one week for transportation to the study site. This discussion implies to the reader that there was some kind of additional uncertainty in the calibration due to the Tekran 2537B unit being powered down. As long as the authors powered up the unit and had argon purge gas flowing for 2-3 days prior to use the permeation system should have re-equilibrated and had no impact on their 2P-LIF system calibration. I suggest the authors clarify the circumstances of the Tekran 2537B operation status. Manual standard addition injections from the Tekran Model 2505 primary calibration source (that the authors indicated they had on site – Line 180) should have been conducted to verify the stability/accuracy of the 2537B instrument perm tube system prior to initiation of the experiment. The authors should report results for any QA/QC injections.

    ii. Lines 122-126: Clarify if the 25' sampling line was heated and shielded from the sun.

    iii. Lines 130-131: If the Tekran 2537B was not able to pull 2 LPM with the additional load of a 25' sampling line it is not clear why the authors simply did not reduce the flow rate set point to 1.5 LPM (the manufacturer recommended flow rate). Adding a supplemental pump to the instrument exhaust and increasing the instruments internal vacuum can impact the permeation tube system performance creating uncertainty in the instruments reported values. The additional vacuum may also have played a role in the permeation tube "malfunction" described by the authors in Lines 230-231. Was the Tekran 2537B checked with manual standard addition injections from the Tekran Model 2505 primary calibration source in the external pump configuration?

    iv. Lines 133-134: The authors indicate that the 2P-LIF system cannot detect RGM, but expressed concern about "deposition of RGM on the sampling lines followed by heterogeneous reduction to GEM". Were any actions taken to filter out RGM from the sampling line (at the manifold port) while allowing the GEM to pass to the instrument like incorporation of a soda and lime trap? Allowing $HgBr_2$ into a Tekran 2537B can have long term contamination effects on the internal components (filter packs, tubing, connectors, and valves).

    v. Lines 148-150: Was a second pump used to maintain flow through the sampling line not being actively sampled by the 2P-LIF system during the TOM difference experiment? Otherwise air in the sampling line void volume not being actively sampled would stagnate and not represent the correct temporal sample duration.

vi. Lines 152-169: How often were the KCl-coated manual annular denuder cleaned and recoated/conditioned?

vii. Lines 231-234: The authors discuss comparison of the UM and UNR Tekran 2537 instruments and bring up the point again about the power down of their 2537B instrument. The authors should clarify how long the system was allowed to stabilize prior to the first perm tube calibration, and how often thereafter it was recalibrated. Was an independent QA auditor part of the study plan to ensure traceability across research group instruments?

b. Line 146: Again PBM is not part of TGM.

c. Line 157: 2003 should be 2002.

d. Line 165: Not heating the KCl-coated annular denuders during sampling can be problematic since the hydrophilic KCl coating will tend to absorb water vapor and can (i) interfere with RGM collection, and (ii) provide surfaces for heterogeneous reactions.

(3) Results:

a. Lines 231-236: In the absence of an independent auditor or standard addition injections to validate the respective instruments perm tube emission rates, it is not possible to definitively establish the reason for the observed differences between the UM and UNR instruments. Based on the described behavior it could be related to contamination of one or both of the instruments with $HgBr_2$ or an unstable permeation tube system. It would be useful for the authors to discuss the observed behavior as a function of the timing of $HgBr_2$ spiking.

b. Lines 300-301: The authors point out in this discussion that the UM 2537B was systematically reporting a higher $Hg^0$ value than the UNR instrument. If the authors believe the divergence between 2537 instruments was due to the UM instrument being turned off for shipment, then the UM instrument would be reporting lower values. This would be due to the fact that the amount of mercury being emitted by the unequilibrated perm tube system would be higher than expected during the calibration since excess $Hg^0$ accumulated on the walls of the perm tube oven would be slowly desorbing – resulting in lower reported ambient concentrations.

c. Line 399: The authors are discussing Fig. 7 in this discussion and then state "In addition, it is clear that the DOHGS system show very different temporal profiles of TOM." I suspect the authors should reference Fig. 9 here since I do not see the DOHGS concentrations presented in Fig. 7.

d. Line 413: I do not see the DOHGS concentrations presented in SI Figure 6.

e. Lines 422-423: I do not see the DOHGS concentrations presented in SI Figures 7-9 as referenced in the text.

f. Lines 447-455: It is unclear what the authors take home message for Section 3.5.3 discussion. It is also unclear why the Spec2 data are shown

in SI Fig. 13 since it was sampling off the manifold while all the other measurements are from the trailer roof.

g. Lines 489-530: Landis et al. 2002 documents the quantitative transport of $HgCl_2$ through the manual denuder elutriator/impactor inlet when properly heated.

h. Lines 490-492: The authors implementation of the manual denuders method described by Landis et al., 2002 deviated in two ways (i) not using the suggested elutriator/impactor inlet to remove large aerosols which may be retained by the denuder causing positive artifacts (the potential contamination of their RGM denuder sample by PBM is later discussed by the authors), and (ii) not heating the denuder system.

i. Lines 493-494: Feng et al., 2004 reference cited by the authors to imply potential loss of RGM by the inlet elutriator/impactor inlet of the manual denuder system does not support their statement. The Feng et al. paper does not use or even mention this system or RGM loss in their particulate mercury methods paper. They used a method described by Lu et al., 1998 that does not use an inlet.

j. Lines 517-522: Manual versus automated denuder and denuder in series experiments described here were previously conducted and presented in Landis et al., 2002. No significance difference between manual and automated systems, and no significant breakthrough from the first denuder. This previous work should be cited.

k. Lines 579-582: The critical review of the experimental design here is warranted and should go further to include recommendations to (i) improve the manifold design and sampling port configurations as previously discussed, (ii) include an independent auditor, and (iii) removal of PM from the gas phase experiments.

---

## Author Response (AR1)

Reply to Referee #2

We appreciate the very comprehensive review of the manuscript. Based on the line numbers that the referee refers to, it appears that it is the original submission was used as the basis for review, rather than the revised submission that it on the website.

To facilitate our response we have set the reviewers comments in red and our response in black type.

General Comments:

(1)In several areas of the paper (e.g., lines 63-64; 144-147) the authors incorrectly include particulate bound mercury (PBM) into their definition of total gaseous mercury (TGM). In the literature TGM is generally used to describe Hg0 + divalent reactive gaseous mercury (RGM). In fact most ambient instruments that quantify TGM (e.g., Tekran Instruments Corporation Model 2537) use an integrated Teflon filter to exclude all particulate matter, and most monitoring networks include an integrated Teflon filter at the inlet of their sampling line to minimize gas/particle interactions in the sampling line that has been shown to create problems in reliably quantifying ambient gaseous mercury species (see below). If the system is pyrolyzing all ambient mercury species to Hg0 for detection, then perhaps coin a new operationally defined term such as total atmospheric mercury (TAM) to avoid confusion.

We appreciate the reviewer picking this up. We have replaced most of the "total gaseous mercury (TGM)" with total mercury (TM) defined as the sum of gaseous elemental and oxidized mercury plus particulate bound mercury. In section 3.2 TGM is the appropriate term.

(2) The authors need to provide additional details on their sampling configuration and calibration procedures (QA/QC) in the methods section to provide readers and reviewers the basic information necessary to inform and judge the implementation of the their 2P-LIF system.

We provided a very detailed description of the instrument as deployed at RAMIX including the sampling and calibration procedures in a recent paper in Atmospheric Measurement Techniques, cited as Bauer et al. (2014).

Bauer, D., Everhart, S., Remeika, J., Tatum Ernest, C., and Hynes, A. J.: Deployment of a Sequential Two-Photon Laser Induced Fluorescence Sensor for the Detection of Gaseous Elemental Mercury at Ambient Levels: Fast, Specific, Ultrasensitive Detection with Parts-Per-Quadrillion Sensitivity, Atmos. Meas. Tech., 7, 4251-4265, www.atmos-meas-tech-discuss.net/7/5651/2014/ doi:10.5194/amtd-7-5651-2014, 2014.

The first sentences of section 2.2 have been rewritten to emphasize this:

"Bauer et al. (2002, 2003, 2014) provide a description of the operating principles of the 2P-LIF instrument. Bauer et al. (2014) provide a detailed description of the instrument deployed at RAMIX including the sampling configurations, data processing, calibration and linearity tests together with examples of experimental data."

 (3) Though not a problem with this paper per se, the design and implementation of the RAMIX study manifold system described by Finley et al. (2013) for this work is problematic and the results described in this paper must be viewed through this lens. The overarching issue was the decision to allow ambient particulate matter (PM1) into the manifold. When the stated objective was to evaluate the efficacy of different measurement methodologies to quantify Hg0 and RGM the logical first step would have been to evaluate these gaseous species first without the complication of spiked gases interacting with ambient particles in the manifold. The curious choice of constructing the manifold using highly porous PFA Teflon tubing also creates multiple potential problems (i) absorption/desorption of spiked gas concentrations following diffusion gradients, (ii) non-conducting material combined with high (187 LPM) flow rate and low relative humidity could lead to electrostatic collection of PM on the internal manifold walls providing additional surfaces for gas phase adsorption, and (iii) poor conduction of heat applied to the external surface of the tubing at such a high manifold flow rate (Finley et al. reported using eight thermocouples to measure external temperature down the length of the manifold but did not report any measuring internal surface or air temperature). Using a blower to maintain flow through the manifold also added a reported 15% uncertainty in the spike concentrations (Finley et al., 2013). All these issues lead to relatively low reported average transmission efficiencies for Hg0 (92%), HgBr2 (76%), and O3 (93%) even under controlled laboratory conditions (Finley et al.,

2013). The range for Hg0 transmission in the manifold was reported to be 71-101% by Prestbo (2014). If the spiked mass of target gases into the manifold were not conserved through the system, then definitive evaluation of the analytical instruments sampling from the manifold were compromised. I agree with the authors that 15% probably underestimates the overall uncertainty for Hg0 (Lines 205-207) and that the system should be characterized as "a semi-quantitative delivery system" (Lines 210-211). If the manifold cannot quantitatively and reproducibly transmit a relatively inert gas like Hg0 there is very little chance of reproducibly transmitting HgBr2.

We have modified "Section 3.1 RAMIX Manifold" because it may give the impression that "The range for Hg0 transmission in the manifold was reported to be 71-101% by Prestbo (2014).", as inferred by the reviewer.

This was not reported by Prestbo but rather is in the Finley et al. paper.

In regards to the issue of particulate matter the Finley et al. paper also contains the following:

"A Teflon coated cyclone inlet (URG 2000-30EA), also heated to 115 °C, was attached to the manifold. The inlet gave a particle size cut of approximately 1 μm at the expected manifold flow rate of 185–230 LPM (manufacturer communication)." This indicates that only the submicron fraction of ambient particulate matter was introduced into the manifold. However we appreciate the comments and note that in "Section 5.0 Future Mercury Intercomparisons:" we discuss the need for sampling in an "unreactive" configuration that would eliminate any issues associated with particulate matter.

We write:

Line 750: Based on the RAMIX results it should consist of a period of direct ambient sampling and then manifold sampling in both reactive and unreactive configurations. For example an unreactive configuration would consist of Hg(0) and oxidized mercury in an $N_2$ diluent eliminating any possibility of manifold reactions and offering the possibility of obtaining a manifold blank response.

(4) This reviewer agrees with the authors that previous RAMIX study papers invoke and discuss mercury oxidation and reduction chemistry that is not supported in either the theoretical or applied literature. Controlling the physical adsorption of gaseous species onto manifold surfaces and aerosols is the logical first direction for which to find answers.

Specific Comments:

(1) Introduction:

a. Lines 49-50: Sprovieri et al., 2016 Atmos. Chem. Phys. Discuss., doi:10.5194/acp-2016-466 is a more contemporary discussion of global background Hg0 concentrations in the Northern and Southern hemispheres from the global GMOS network.

We had replaced Sprovieri et al. (2010) with Slemr et al., 2011, in the revised submission.  We have added  Sprovieri et al. (2016) as an additional reference.

b. Lines 63-64: PBM is not part of TGM.

Modified as described above (2) Experimental:

As indicated above, the authors need to supplement the QA/QC information.

See reply to point 2) in general comments.

i. Lines 122-126: The authors discuss calibrating the 2P-LIF system using a Tekran 2537B as a secondary transfer standard, and that unlike their previous field work the unit was turned off for one week for transportation to the study site. This discussion implies to the reader that there was some kind of additional uncertainty in the calibration due to the Tekran 2537B unit being powered down. As long as the authors powered up the unit and had argon purge gas flowing for 2-3 days prior to use the permeation system should have re-equilibrated and had no impact on their 2P-LIF system calibration. I suggest the authors clarify the circumstances of the Tekran 2537B operation status. Manual standard addition injections from the Tekran Model 2505 primary calibration source (that the authors indicated they had on site – Line 180) should have been conducted to verify the stability/accuracy of the 2537B instrument perm tube system prior to initiation of the experiment. The authors should report results for any QA/QC injections.

In the initial period of operation we saw instability in the response factors from calibrations and the overall response between calibrations. This is discussed below. However we cannot clearly identify these problems as being due to the fact that the Tekran 2537 was shipped and went without power for one week and we have removed that statement from section 2.2.  Section 3.2 now includes the statement: "Problems with instability in the UM Tekran may have been associated with the use of an external pump to supplement the internal Tekran pump, or with the fact that the UM instrument had been powered down for almost one week and relocated to a site at a significantly different ambient pressure."

The Tekran 2537 was calibrated using the internal permeation source. The Tekran Model 2505 primary calibration source was at a different location at the University of Nevada, Reno. This was the location of the LIF system that was used for denuder analysis. In retrospect it is evident that we should have performed additional  QA/QC procedures but we would note that at this period, the start of the intercomparison we were primarily focused on assembling two completely different LIF systems, the 2P-LIF system at the RAMIX site and the LIF system for denuder analysis that was located in a laboratory at UNR. We have modified the text in "3.2 UM Tekran Performance " to discuss this as detailed below.

ii. Lines 122-126: Clarify if the 25' sampling line was heated and shielded from the sun.

The sampling line was not heated and was not shielded from the sun and this is now noted in the manuscript.

iii. Lines 130-131: If the Tekran 2537B was not able to pull 2 LPM with the additional load of a 25' sampling line it is not clear why the authors simply did not reduce the flow rate set point to 1.5 LPM (the manufacturer recommended flow rate). Adding a supplemental pump to the instrument exhaust and increasing the instruments internal vacuum can impact the permeation tube system performance creating uncertainty in the instruments reported values. The additional vacuum may also have played a role in the permeation tube "malfunction" described by the authors in Lines 230-231. Was the Tekran 2537B checked with manual standard addition injections from the Tekran Model 2505 primary calibration source in the external pump configuration?

There is an error in the manuscript here and in Bauer et al. (2014) and this is now corrected. The Tekran was indeed sampling at 1.5 LPM but it was necessary to add a supplemental pump to be able to sample at this flow. We were unaware that this might present potential problems.  As noted above  the  Tekran Model 2505 primary calibration source was at a different location at the University of Nevada, Reno and it was not used to check the Tekran 2537 calibration.

iv. Lines 133-134: The authors indicate that the 2P-LIF system cannot detect RGM, but expressed concern about "deposition of RGM on the sampling lines followed by

heterogeneous reduction to GEM". Were any actions taken to filter out RGM from the sampling line (at the manifold port) while allowing the GEM to pass to the instrument like incorporation of a soda and lime trap? Allowing HgBr2 into a Tekran 2537B can have long term contamination effects on the internal components (filter packs, tubing, connectors, and valves).

No actions were taken to filter out RGM from the sample line. This is now explicitly stated in the manuscript.

v. Lines 148-150: Was a second pump used to maintain flow through the sampling line not being actively sampled by the 2P-LIF system during the TOM difference experiment? Otherwise air in the sampling line void volume not being actively sampled would stagnate and not represent the correct temporal sample duration.

Both lines were continuously sampled at 10 L/min and the flow to the fluorescence cell was switched between the lines. This is now explicitly stated in the manuscript.

vi. Lines 152-169: How often were the KCl-coated manual annular denuder cleaned and recoated/conditioned?

The denuders were cleaned and recoated prior to the RAMIX deployment. Prior to sampling, the denuders were cleaned by heating to 500 °C and then bagged and taken to the sampling site. This is now explicitly stated in the text.

vii. Lines 231-234: The authors discuss comparison of the UM and UNR Tekran 2537 instruments and bring up the point again about the power down of their 2537B instrument. The authors should clarify how long the system was allowed to stabilize prior to the first perm tube calibration, and how often thereafter it was recalibrated. Was an independent QA auditor part of the study plan to ensure traceability across research group instruments?

See below b. Line 146: Again PBM is not part of TGM.

Modified as described above c. Line 157: 2003 should be 2002.

Corrected d. Line 165: Not heating the KCl-coated annular denuders during sampling can be problematic since the hydrophilic KCl coating will tend to absorb water vapor and can (i) interfere with RGM collection, and (ii) provide surfaces for heterogeneous reactions.

We would note that the humidity in Reno was low during the RAMIX experiment.

(3) Results:

a. Lines 231-236: In the absence of an independent auditor or standard addition injections to validate the respective instruments perm tube emission rates, it is not possible to definitively establish the reason for the observed differences between the UM and UNR instruments. Based on the described behavior it could be related to contamination of one or both of the instruments with HgBr2 or an unstable permeation tube system. It would be useful for the authors to discuss the observed behavior as a function of the timing of HgBr2 spiking.

b. Lines 300-301: The authors point out in this discussion that the UM 2537B was systematically reporting a higher Hg0 value than the UNR instrument. If the authors believe the divergence between 2537 instruments was due to the UM instrument being turned off for shipment, then the UM instrument would be reporting lower values. This would be due to the fact that the amount of mercury being emitted by the unequilibrated perm tube system would be higher than expected during the calibration since excess Hg0 accumulated on the walls of the perm tube oven would be slowly desorbing – resulting in lower reported ambient concentrations.

See below c. Line 399: The authors are discussing Fig. 7 in this discussion and then state "In addition, it is clear that the DOHGS system show very different temporal profiles of TOM." I suspect the authors should reference Fig. 9 here since I do not see the DOHGS concentrations presented in Fig. 7.

This has been clarified.

d. Line 413: I do not see the DOHGS concentrations presented in SI Figure 6.

The DOHGS concentrations are shown in Fig. 10. The text has been modified.

e. Lines 422-423: I do not see the DOHGS concentrations presented in SI Figures 7-9 as referenced in the text.

As noted above, the DOHGS concentrations are shown in Fig. 10. The text has been modified.

f. Lines 447-455: It is unclear what the authors take home message for Section 3.5.3 discussion. It is also unclear why the Spec2 data are shown in SI Fig. 13 since it was sampling off the manifold while all the other measurements are from the trailer roof.

SI Fig. 13 shows both Spec 1 and Spec 2 data. Spec 2 was sampling ambient air but Spec 1 was sampling from the manifold which was spiked with $HgBr_2$. The Spec 1 data should not have been included and it has been removed.

g. Lines 489-530: Landis et al. 2002 documents the quantitative transport of HgCl2 through the manual denuder elutriator/impactor inlet when properly heated.

See response below to i)

h. Lines 490-492: The authors implementation of the manual denuders method described by Landis et al., 2002 deviated in two ways (i) not using the suggested elutriator/impactor inlet to remove large aerosols which may be retained by the denuder causing positive artifacts (the potential contamination of their RGM denuder sample by PBM is later discussed by the authors), and (ii) not heating the denuder system.

Now reads "As we describe above, our use of manual denuders was similar to that described by Landis et al. (2002) with the exception that we did not incorporate the integrated elutriator/acceleration jet and impactor/coupler on the denuder inlet and the denuders were not heated. "

i) Lines 493-494: Feng et al., 2004 reference cited by the authors to imply potential loss of RGM by the inlet elutriator/impactor inlet of the manual denuder system does not support their statement. The Feng et al. paper does not use or even mention this system or RGM loss in their particulate mercury methods paper. They used a method described by Lu et al., 1998 that does not use an inlet.

This reference has been corrected to Feng et al. (2003).

Feng, X.; Lu, J.Y.; Hao, Y.; Banic, C.; Schroeder, W. H. : Evaluation and application of a gaseous mercuric chloride source , Anal. Bioanal. Chem., 376, 1137-1140, 2003.

The abstract includes the following:

"It is shown that, under the experimental conditions examined, KCl-coated annular quartz denuders designed for ambient reactive gaseous mercury (RGM) collection could quantitatively collect $HgCl_2$. It is also demonstrated that the impactors used to remove coarse airborne particulate matter could lead to a loss of up to one third of the HgCl2 in the gas stream."

The authors cite Landis et al. (2002) but do not state that the impactors and denuders were heated.

Our text has been modified as follows:

"Landis et al. (2002) suggest that $HgCl_2$ is quantitatively transported through the manual denuder elutriator/impactor inlet when properly heated. In later work Feng et al. (2003) reported that such impactors could reduce the efficiency of RGM collection although in that work there is no reference to the temperature of the impactor. In this work no type of particle filtering was used on the inlets."

j. Lines 517-522: Manual versus automated denuder and denuder in series experiments described here were previously conducted and presented in Landis et al., 2002. No significance difference between manual and automated systems, and no significant breakthrough from the first denuder. This previous work should be cited.

The Feng et al. (2003) also included laboratory experiments on manual denuders in series. Both works are now cited.

k. Lines 579-582: The critical review of the experimental design here is warranted and should go further to include recommendations to (i) improve the manifold design and sampling port configurations as previously discussed, (ii) include an independent auditor, and (iii) removal of PM from the gas phase experiments.

We have not tried to provide a detailed recommendation on the future manifold design. The reference to the GASIE campaign that included oversight by an independent panel consisting of three atmospheric scientists none of whom were involved in $SO_2$ research and our suggestion that:

"We would suggest that a future mercury intercomparison should be blind with independent oversight."

Imply the inclusion of independent auditors.

As we have noted above the RAMIX manifold did incorporate a Teflon coated cyclone inlet.

i. Lines 122-126: The authors discuss calibrating the 2P-LIF system using a Tekran 2537B as a secondary transfer standard, ..........

We have provided a single response to the comments related to the operation of the UM Tekran vii. Lines 231-234: The authors discuss comparison of the UM and UNR Tekran 2537 instruments .....................

(3) Results:

a. Lines 231-236: In the absence of an independent auditor or standard addition injections ..................

b. Lines 300-301: The authors point out in this discussion that the UM 2537B was systematically reporting a higher Hg0 value than the UNR instrument. ................

As we have noted above the start of the intercomparison we were primarily focused on assembling two completely different LIF systems, the 2P-LIF system at the

RAMIX site and the LIF system for denuder analysis that was located in a laboratory at UNR. We should also note that we are not Tekran experts and the UM Tekran was not part of the primary intercomparison, in retrospect it is evident that we should have performed additional  QA/QC procedures on the Tekran. We believe the key point here is that it was not reasonable for Gustin et al. to use this data to conclude that by some unidentified mechanism oxidized mercury was being converted to

Hg(0) and that the UM Tekran was measuring total mercury. It is also noteworthy that the offsets shown in Fig. 1 occurred prior to the start of the manifold spikes of

$HgBr_2$ and cannot be associated with the elevated levels of $HgBr_2$ that were introduced into the manifold on Sept. 5th. We have now modified section 2.2 and added a section in Supplementary Information giving more details about the calibration and response instabilities in the UM Tekran.

We have modified Section 2.2 as follows:

**"3.2 UM Tekran Performance**

In evaluating the first week of the UM RAMIX measurements it became clear that there was some non-linearity in the relative responses of the 2P-LIF and UM Tekran systems and that better agreement was obtained by referencing the Hg(0)

concentration to the UNR Tekran. Gustin et al., (2013) concluded that the UNR

Tekran, based on the inlet configuration, only measured Hg(0) and they suggested that the UM system, due to the long sampling line, was measuring total gaseous mercury (TGM). We compared the manifold Hg(0) readings from the UM and UNR

Tekrans over the first 260 hours in which we took measurements. The absolute concentration difference relative to the UNR instrument is shown in Figure 1. Hour zero corresponds to 9 am on August 26th when we started measurements and hour corresponds to midnight on September 5th. Over the first 24 hours the UM

Tekran is offset by ~0.5 ng m$^{-3}$ and then jumps to ~ 2 ng m$^{-3}$ at hour 30 on August

27th with the difference decreasing over the next week of measurements in an almost linear fashion. Over most of this period the UW Tekran did not report Hg(0)

measurements other than a small set of measurements on August 28th that are offset by ~0.5 ng m$^{-3}$ relative to the UNR Tekran. It can be seen that by hour 250 on

September 5th all three instruments had converged. After this period the agreement between the UW, UNR and UM Tekrans was good until September 8th, when the UM

instrument became contaminated after a malfunction of our permeation oven, requiring replacement with a backup Tekran 2537A unit. Both the absolute response and the response factor, i.e. the calibration factor of the UM Tekran were somewhat unstable during this period and additional details are provided in the

Supplementary Information. Our focus during this initial period of the intercomparison was on the two laser / sampling systems that were being set up. In retrospect we can acknowledge that greater attention should have been paid to quality assurance with the UM Tekran. We conclude that the difference between the

UM and UNR instruments is an experimental artifact. Problems with instability in the UM Tekran may have been associated with the use of an external pump to supplement the internal Tekran pump, or with the fact that the UM instrument had been powered down for almost one week and relocated to a site at a significantly different ambient pressure. It is also noteworthy that the offsets shown in Fig. 1 occurred prior to the start of the manifold spikes of $HgBr_2$ and cannot be associated with the elevated levels of $HgBr_2$ that were introduced into the manifold on Sept. 5th. The observations cannot, in our view, be indicative of any type of chemistry within the manifold, nor can it be indicative of the UM instrument measuring TGM rather than Hg(0). "

The Supplementary Information now contains the following:

Initial operation of the UM Tekran.

The UM Tekran was powered up after arrival at the RAMIX site and calibrated after 24 hours and again after another 24 hours. The response factors were consistent at $\sim 6 \times 10^7$. During this period the Tekran was sampling ambient air and was not connected to the RAMIX manifold which, as noted in the main text, was below ambient pressure. After connection to the manifold via 25 ft of 0.25 inch tubing an external pump was connected to allow the instrument to maintain a 1.5 L min$^{-1}$ sampling rate. At hour 72 on August 29th the Tekran was recalibrated and the response factor dropped to $\sim 4.3 \times 10^7$. At the next recalibration on September 2nd at hour 175 the response factor increased to $\sim 6 \times 10^7$ and then on September 7th decreased to $\sim 4.5 \times 10^7$. In addition to the instability in the response factors there was some instability in overall response that was not related to calibration. After sampling began from the manifold at hour zero, which corresponds to 9 am on August 26th, the UM Tekran was offset by $\sim +0.5$ ng m$^{-3}$ with respect to the UNR

Tekran and this offset increased to ~ +2 ng m$^{-3}$ at hour 30 on August 27$^{th}$. This increase in offset occurred after the UM system was disconnected from the manifold and flushed with the N$_2$ blowoff from a liquid nitrogen tank that gave good zeros but would have increased the pressure in the system to ambient pressure. The large offset is caused by the UM instrument reading high and the UNR instrument reading low and then over the next 45 hours the instruments converge. This instability occurred prior to any manifold spiking with HgBr$_2$ As noted in Section 3.2 of the manuscript, 
[revised manuscript text omitted]